# TFG: Unified Training-Free Guidance for Diffusion Models

**Haotian Ye**[1*]    **Haowei Lin**[2*]    **Jiaqi Han**[1*]    **Minkai Xu**[1]    **Sheng Liu**[1]
**Yitao Liang**[2]    **Jianzhu Ma**[3]    **James Zou**[1]
**Stefano Ermon**[1]
[1]Stanford University    [2]Peking University    [3]Tsinghua University

## Abstract

Given an unconditional diffusion model and a predictor for a target property of interest (*e.g.*, a classifier), the goal of training-free guidance is to generate samples with desirable target properties without additional training. Existing methods, though effective in various individual applications, often lack theoretical grounding and rigorous testing on extensive benchmarks. As a result, they could even fail on simple tasks, and applying them to a new problem becomes unavoidably difficult. This paper introduces a novel algorithmic framework encompassing existing methods as special cases, unifying the study of training-free guidance into the analysis of an algorithm-agnostic design space. Via theoretical and empirical investigation, we propose an efficient and effective hyper-parameter searching strategy that can be readily applied to any downstream task. We systematically benchmark across 7 diffusion models on 16 tasks with 40 targets, and improve performance by 8.5% on average. Our framework and benchmark offer a solid foundation for conditional generation in a training-free manner.[1]

## 1  Introduction

Recent advancements in generative models, particularly diffusion models [61, 21, 62, 66], have demonstrated remarkable effectiveness across vision [65, 48, 52], small molecules [74, 73, 24], proteins[1, 72], audio [35, 29], 3D objects [40, 41], and many more. Diffusion models estimate the gradient of log density (i.e., Stein score, [67]) of the data distribution [65] via denoising learning objectives, and can generate new samples via an iterative denoising process. With impressive scalability to billions of data [58], future diffusion models have the potential to serve as *foundational* generative models across a wide range of applications. Consequently, the problem of conditional generation based on these models, *i.e.*, tailoring outputs to satisfy user-defined criteria such as labels, attributes, energies, and spatial-temporal information, is becoming increasingly important [63, 2].

Conditional generation methods like classifier-based guidance [66, 7] and classifier-free guidance [23] typically require training a specialized model for each conditioning signal (e.g., a noise-conditional classifier or a text-conditional denoiser). This resource-intensive and time-consuming process greatly limits their applicability. In contrast, *training-free guidance* aims to generate samples that align with certain targets specified through an *off-the-shelf* differentiable target predictor without involving any additional training. Here, a target predictor can be any classifier, loss function, probability function, or energy function used to score the quality of the generated samples.

In classifier-based guidance [66, 7], where a noise-conditional classifier is specifically trained to predict the target property on both clean and noisy samples, incorporating guidance in the diffusion

---

*Equal contribution. Corresponding to mailto:haotianye@stanford.edu.

[1]Code is available at https://github.com/YWolfeee/Training-Free-Guidance.

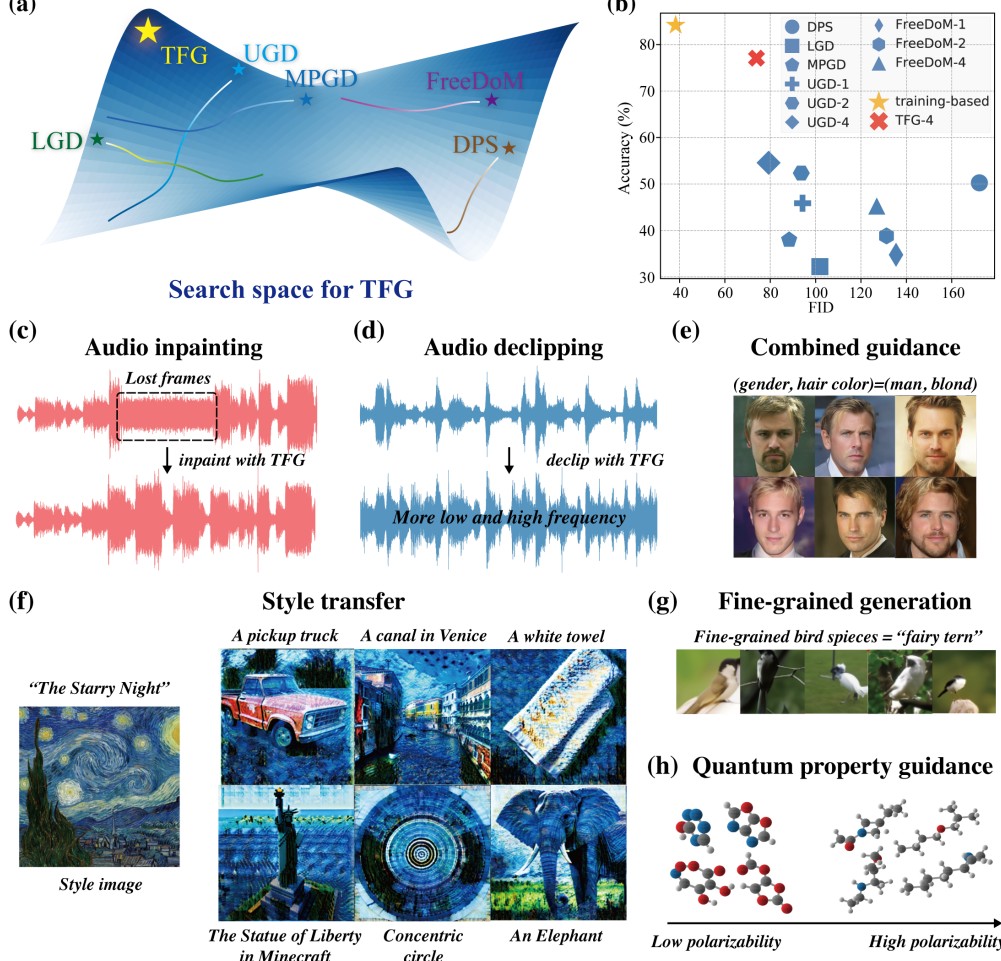

Figure 1: **(a)** Illustration of the unified search space of our proposed TFG, where the height (color) stands for performance. Existing algorithms search along sub-manifolds, while TFG results in improved guidance thanks to its extended search space. **(b)** The label accuracy (higher the better) and Fréchet inception distance (FID, lower the better) of different methods for the label guidance task on CIFAR10 [30], averaged across ten labels. Ours (TFG-4) performs much closer to training-based methods. **(c∼h)** TFG generated samples across various tasks in vision, audio, and geometry domains.

process is straightforward since the gradient of the classifier is an unbiased driving term. Training-free guidance, however, is fundamentally more difficult. The primary challenge lies in leveraging a target predictor trained solely on *clean* samples to offer guidance on *noisy* samples. Although various approaches have been proposed [18, 63, 6, 2, 78] and are effective for some individual tasks, theoretical grounding and comprehensive benchmarks are still missing. Indeed, existing methods fail to produce satisfactory samples for label guidance even on simple datasets such as CIFAR10 (Figure 1). Moreover, the lack of quantitative comparisons between these methods makes it difficult for practitioners to identify an appropriate algorithm for a new application scenario.

This paper proposes a novel and general algorithmic framework for (and also named as) **Training Free Guidance** (TFG). We show that existing approaches are special cases of the TFG as they correspond to particular hyper-parameter subspace in our unified space. In other words, TFG naturally simplifies and reduces the study of training-free guidance, as well as the comparisons between existing methods, into the analysis of hyper-parameter choices in our unified design space. Within our framework, we analyze the underlying theoretical motivation of each hyper-parameter and conduct comprehensive experiments to identify their influence. Our systematic study offers novel insights into the principles behind training-free guidance, allowing for a transparent and efficient survey of the problem.

Based on the framework, we propose a hyper-parameter searching strategy for general downstream tasks. We comprehensively benchmark TFG and existing algorithms across 16 tasks (ranging from images to molecules) and 40 targets. TFG achieves superior performance across all datasets, outperforming existing methods by 8.5% on average. In particular, it excels in generating user-required samples in various scenarios, regardless of the complexity of targets and datasets.

In summary, we (1) propose TFG that unifies existing algorithms into a design space, (2) theoretically and empirically analyze the space to propose an effective space-searching strategy for general problems, and (3) benchmark all methods on numerous qualitatively different tasks to present the superiority of TFG and the guideline for future research in training-free conditional generation algorithms. This advancement demonstrates the efficacy of TFG and establishes a robust and comprehensive benchmark for future research in training-free conditional generation algorithms.

## 2 Background

**Generative diffusion model.** A generative diffusion model is a neural network that can be used to sample from an unconditional distribution $p_0(\boldsymbol{x})$ with the support on any continuous sample space $\mathcal{X}$ [21, 62, 64, 27]. For instance, $\mathcal{X}$ could be $[-1, 1]^{d \times d \times 3}$ representing the RGB colors of $d \times d$ images [4, 22], or $\mathbb{R}^{3d}$ representing the 3D coordinates of molecules with $d$ atoms [24, 74, 73]. Given a data $\boldsymbol{x}_0$ sampled from $p_0(\boldsymbol{x})$, a time step $t \in [T] \triangleq \{1, \cdots, T\}$, a corresponding noisy datapoint is constructed as $\boldsymbol{x}_t = \sqrt{\bar{\alpha}_t}\boldsymbol{x}_0 + \sqrt{1 - \bar{\alpha}_t}\epsilon$ where $\epsilon \sim \mathcal{N}(\boldsymbol{0}, \boldsymbol{I})$ and $\{\bar{\alpha}_t\}_{t=1}^T$ is a set of pre-defined monotonically decreasing parameters used to control the noise level. Following [21], we further define $\alpha_t = \bar{\alpha}_t/\bar{\alpha}_{t-1}$ for $t > 1$ and $\alpha_1 = \bar{\alpha}_1$. The diffusion model $\epsilon_\theta : \mathcal{X} \times [T] \mapsto \mathcal{X}$ parameterized by $\theta$ is trained to predict the noise $\epsilon$ that was added on $\boldsymbol{x}_t$ with p.d.f $p_t(\boldsymbol{x}_t) = \int_{\boldsymbol{x}_0} p_0(\boldsymbol{x}_0)p_{t|0}(\boldsymbol{x}_t|\boldsymbol{x}_0)\mathrm{d}\boldsymbol{x}_0$[2]. In theory, this corresponds to learning the score of $p_t(\boldsymbol{x})$ [65], i.e.,

$$\arg\min_{\epsilon_\theta} \sum_{t=1}^T \mathbb{E}_{\boldsymbol{x}_0 \sim p_0(\boldsymbol{x}_0), \epsilon \sim \mathcal{N}(\boldsymbol{0}, \boldsymbol{I})} \|\epsilon_\theta(\boldsymbol{x}_t, t) - \epsilon\| = -\sqrt{1 - \bar{\alpha}_t}\nabla \log p_t. \tag{1}$$

For sampling, we start from $\boldsymbol{x}_T \sim \mathcal{N}(\boldsymbol{0}, \boldsymbol{I})$ and gradually sample $\boldsymbol{x}_{t-1} \sim p_{t-1|t}(\boldsymbol{x}_{t-1}|\boldsymbol{x}_t)$. This conditional probability is not directly computable, and in practice, DDIM [62] samples $\boldsymbol{x}_{t-1}$ via

$$\boldsymbol{x}_{t-1} = \sqrt{\bar{\alpha}_{t-1}}\boldsymbol{x}_{0|t} + \sqrt{1 - \bar{\alpha}_{t-1} - \sigma_t^2}\frac{\boldsymbol{x}_t - \sqrt{\bar{\alpha}_t}\boldsymbol{x}_{0|t}}{\sqrt{1 - \bar{\alpha}_t}} + \sigma_t\epsilon, \tag{2}$$

where $\{\sigma_t\}_{t=1}^T$ are DDIM parameters, $\epsilon \sim \mathcal{N}(\boldsymbol{0}, \boldsymbol{I})$, and

$$\boldsymbol{x}_{0|t} = m(\boldsymbol{x}_t) \triangleq \frac{\boldsymbol{x}_t - \sqrt{1 - \bar{\alpha}_t}\epsilon_\theta(\boldsymbol{x}_t, t)}{\sqrt{\bar{\alpha}_t}} \tag{3}$$

is the predicted sample given $\boldsymbol{x}_t$. According to the Tweedie's formula [11, 51], $\boldsymbol{x}_{0|t}$ equals to the conditional expectation $\mathbb{E}[\boldsymbol{x}_0|\boldsymbol{x}_t]$ under perfect optimization of $\epsilon_\theta$ in Equation (1). It has been theoretically established that the above sampling process results in $\boldsymbol{x}_0 \sim p(\boldsymbol{x})$ under certain assumptions.

**Target predictor.** For a user required target $c$, we use a predictor $f_c(\boldsymbol{x}) : \mathcal{X} \mapsto \mathbb{R}_+ \cup \{0\}$ [3] to represent how well a sample $\boldsymbol{x}$ is aligned with the target (higher the better). Here $f_c(x)$ can be a conditional probability $p_0(c|\boldsymbol{x})$ for a label $c$ [62, 14], a Boltzmann distribution $\exp^{-e_c(\boldsymbol{x})}$ for any pre-defined energy function $e_c$ [31, 63, 38], the similarity of two features [47], or even their combinations. The goal is to samples from the conditional distribution

$$p_0(\boldsymbol{x}|c) \triangleq \frac{p_0(\boldsymbol{x})f_c(\boldsymbol{x})}{\int_{\tilde{\boldsymbol{x}}} p_0(\tilde{\boldsymbol{x}})f_c(\tilde{\boldsymbol{x}})\mathrm{d}\tilde{\boldsymbol{x}}}. \tag{4}$$

---

[2]In this paper, we use $p(\boldsymbol{x})$ to represent the probability density function (p.d.f.), and $p_t(\boldsymbol{x}), p_{t|s}(\boldsymbol{x}|\tilde{\boldsymbol{x}})$ to represent the probability at time step $t$ and the conditional probability of $\boldsymbol{x}$ at time step $t$ given $\tilde{\boldsymbol{x}}$ at time step $s$.

[3]Here $c$ can has any mathematical form. We assume in this paper that $f_c(\boldsymbol{x})$ has finite two norm, i.e. $\int_{\boldsymbol{x} \in \mathcal{X}}[f_c^2(\boldsymbol{x})] < +\infty$, such that the probabilistic explanation is well-defined.

**Training-based guidance for diffusion models.** [66] proposes to train a time-dependent classifier to fit $f_c(\boldsymbol{x}_t, t) \triangleq \mathbb{E}_{\boldsymbol{x}_0 \sim p_{0|t}(\cdot|\boldsymbol{x}_t)} f_c(\boldsymbol{x}_0)$. This can be regarded as a predictor over noisy samples. Since

$$\begin{aligned}
\nabla_{\boldsymbol{x}_t} \log p_t(\boldsymbol{x}_t|c) &= \nabla_{\boldsymbol{x}_t} \log \int_{\boldsymbol{x}_0} p_{t|0}(\boldsymbol{x}_t|\boldsymbol{x}_0) p_0(\boldsymbol{x}_0|c) \mathrm{d}\boldsymbol{x}_0 \\
&= \nabla_{\boldsymbol{x}_t} \log \int_{\boldsymbol{x}_0} p_t(\boldsymbol{x}_t) p_{0|t}(\boldsymbol{x}_0|\boldsymbol{x}_t) f_c(\boldsymbol{x}_0) \mathrm{d}\boldsymbol{x}_0 \\
&= \nabla_{\boldsymbol{x}_t} \log p_t(\boldsymbol{x}_t) + \nabla_{\boldsymbol{x}_t} \log f_c(\boldsymbol{x}_t, t),
\end{aligned} \tag{5}$$

if we denote the trained classifier as $f(\boldsymbol{x}_t)$ (that implicitly depends on $c$ and model parameters), we can replace $\epsilon_\theta(\boldsymbol{x}_t, t)$ in Equation (3) by $\epsilon_\theta(\boldsymbol{x}_t, t) - \sqrt{1 - \bar{\alpha}_t} \nabla_{\boldsymbol{x}_t} \log f(\boldsymbol{x}_t)$ upon sampling to obtain unbiased sample $\boldsymbol{x}_0 \sim p_0(\boldsymbol{x}_0|c)$. On the other hand, [23] proposes the classifier-free diffusion guidance approach. Instead of training a time-dependent predictor $f$, it encodes conditions $c$ directly into the diffusion model as $\epsilon_\theta(\boldsymbol{x}, c, t)$ and trains this condition-aware diffusion model with sample-condition pairs. Both methods have been proven effective when training resources are available.

This paper in contrast focuses on conditional generation in a *training-free* manner: given a diffusion model $\epsilon_\theta(\boldsymbol{x}, t)$ and an off-the-shelf target predictor $f(\boldsymbol{x})$ (we omit the subscript $c$ below), we aim to generate samples from $p_0(\boldsymbol{x}|c)$ without any additional training. Unlike training-based methods that can accurately estimate $f(\boldsymbol{x}_t, t)$, training-free guidance is significantly more difficult since it involves guiding a noisy data $\boldsymbol{x}_t$ using $f(\boldsymbol{x})$ defined over the *clean* data space.

## 2.1 Existing algorithms

Most existing methods take advantage of the predicted sample $\boldsymbol{x}_{0|t}$ defined in Equation (3) and use the gradient of $f(\boldsymbol{x})$ for guidance. We review and summarize five existing approaches below, and provide a schematic and a copy of pseudo-code in Appendix B for the sake of reference. Due to the variety in underlying intuitions and implementations, coupled with a lack of quantitative comparisons among these methods, it is challenging to discern which operations are crucial and which are superfluous, a problem we address in Section 3.

**DPS** [6] was initially proposed to solve general noisy inverse problems for image generation: for a given condition $\boldsymbol{y}$ and a transformation operator $\mathcal{A}$, we aim to generate image $\boldsymbol{x}$ such that $\|\mathcal{A}(\boldsymbol{x}) - \boldsymbol{y}\|_2$ is small. For instance, in super-resolution task [71], the operator $\mathcal{A}$ is a down-sampling operator, and $\boldsymbol{y}$ is a low-resolution image. DPS replaces $\nabla \log f(\boldsymbol{x}_t, t)$ in Equation (5) by $\nabla_{\boldsymbol{x}_t} \log f(m(\boldsymbol{x}_t))$. As suggested in [63], this corresponds to a point estimation of the conditional density $p_{0|t}(\boldsymbol{x}_0|\boldsymbol{x}_t)$.

**LGD** [63] replaces the point estimation in DPS and proposes to estimate $f(\boldsymbol{x}_t, t)$ with a Gaussian kernel $\mathbb{E}_{\boldsymbol{x} \sim \mathcal{N}(\boldsymbol{x}_{0|t}, \sigma_t^2 \boldsymbol{I})} f(\boldsymbol{x}, t)$, where the expectation is computed using Monte-Carlo sampling [59].

**FreeDoM** [78] generalizes DPS by introducing a "recurrent strategy" (called "time-travel strategy" [39, 10, 70]) that iteratively denoises $\boldsymbol{x}_{t-1}$ from $\boldsymbol{x}_t$ and adds noise to $\boldsymbol{x}_{t-1}$ to regenerate $\boldsymbol{x}_t$ back and forth. This strategy empirically enhances the strength of the guidance at the cost of additional computation. FreeDoM also points out the importance of altering guidance strength at different time steps $t$, but a comprehensive study on which schedule is better is not provided.

**MPGD** [18] is proposed for manifold preserving tasks, e.g., the target predictor is supposed to generate samples on a given manifold. It computes the gradient of $\log f(\boldsymbol{x}_{0|t})$ to $\boldsymbol{x}_{0|t}$ instead of $\boldsymbol{x}_t$, i.e., $\nabla_{\boldsymbol{x}_{0|t}} \log f(\boldsymbol{x}_{0|t})$ to avoid the back-propagation through the diffusion model $\epsilon_\theta$ that is highly inefficient. This strategy is effective in manifold-preserving problems, but whether it can be generalized to general training-free problems is unclear. In addition to the computation difference, theoretical understanding on the difference between gradients to $\boldsymbol{x}_{0|t}$ and $\boldsymbol{x}_t$ is missing.[4]

**UGD** [2] builds on FreeDoM, with the difference that it additionally solves a backward optimization problem $\Delta_0 = \arg\max_\Delta f(\boldsymbol{x}_{0|t} + \Delta)$ and guides $\boldsymbol{x}_{0|t}$ and $\boldsymbol{x}_t$ simultaneously. UGD also implements the "recurrent strategy" to further improve generation quality.

---

[4]MPGD additionally proposed to use an auto-encoder to improve the quality of $\boldsymbol{x}_{0|t}$. However, an auto-encoder is usually inaccessible or requires training; thus, we don't apply it in our training-free scenario.

# 3 TFG: A Unified Framework for Training-free Guidance

Despite the array of algorithms available and their reported successes in various applications, we conduct a case study on CIFAR10 [30] to illustrate the challenging nature of training-free guidance and the insufficiency of existing methods. Specifically, for each of the ten labels, we use the pretrained diffusion model and classifiers from [7, 9] to generate 2048 samples, where the hyper-parameters are selected via a grid search for the fairness of comparison. We compute the FID and the label accuracy evaluated by another classifier [20] and present results in Figure 1. Even in such a relatively simple setting, all training-free approaches significantly underperform training-based guidance, with a significant portion of generated images being highly unnatural (when guidance is strong) or irrelevant to the label (when guidance is weak). These findings reveal the fundamental challenges and highlight the necessity of a comprehensive study. Unfortunately, comparisons and analyses of existing approaches are missing or primarily qualitative, limiting deeper investigation in this field.

## 3.1 Unification and extension

This sections introduces our unified framework for training-free guidance (TFG, Algorithm 1) and formally defines its design space in Definition 3.1. We demonstrate the advantage of TFG by drawing connections between TFG and other algorithms to show that existing algorithms are encompassed as special cases. Based on this, *all comparisons and studies of training-free algorithms automatically become the study within the hyper-parameter space of our framework*. This allows us to analyze the techniques theoretically and empirically, and choose an appropriate hyper-parameter for a specific downstream task efficiently and effectively, as shown in Section 4.

---

**Algorithm 1** Training-Free Guidance

1: **Input:** Unconditional diffusion model $\epsilon_\theta$, target predictor $f$, guidance strength $\boldsymbol{\rho}, \boldsymbol{\mu}, \bar{\gamma}$, number of steps $T, N_{\text{recur}}, N_{\text{iter}}$
2: $\boldsymbol{x}_T \sim \mathcal{N}(\boldsymbol{0}, \boldsymbol{I})$
3: **for** $t = T, \cdots, 1$ **do**
4:     Define function $\tilde{f}(\boldsymbol{x}) = \mathbb{E}_{\delta \sim \mathcal{N}(\boldsymbol{0}, \boldsymbol{I})} f(\boldsymbol{x} + \bar{\gamma}\sqrt{1 - \bar{\alpha}_t}\delta)$
5:     **for** $r = 1, \cdots, N_{\text{recur}}$ **do**
6:         $\boldsymbol{x}_{0|t} = (\boldsymbol{x}_t - \sqrt{1 - \bar{\alpha}_t}\epsilon_\theta(\boldsymbol{x}_t, t))/\sqrt{\bar{\alpha}_t}$              ▷ Obtain the predicted data
7:         $\Delta_t = \rho_t \nabla_{\boldsymbol{x}_t} \log \tilde{f}(\boldsymbol{x}_{0|t})$
8:         $\Delta_0 = \Delta_0 + \mu_t \nabla_{\boldsymbol{x}_{0|t}} \log \tilde{f}(\boldsymbol{x}_{0|t} + \Delta_0)$     ▷ Iterate $N_{\text{iter}}$ times starting from $\Delta_0 = \boldsymbol{0}$
9:         $\boldsymbol{x}_{t-1} = \text{Sample}(\boldsymbol{x}_t, \boldsymbol{x}_{0|t}, t) + \Delta_t/\sqrt{\bar{\alpha}_t} + \sqrt{\bar{\alpha}_{t-1}}\Delta_0$     ▷ Sample follows Equation (2)
10:         $\boldsymbol{x}_t \sim \mathcal{N}(\sqrt{\alpha_t}\boldsymbol{x}_{t-1}, \sqrt{1 - \alpha_t}\boldsymbol{I})$                ▷ Recurrent strategy
11:     **end for**
12: **end for**
13: **Output:** Conditional sample $\boldsymbol{x}_0$

---

**Definition 3.1.** Given a denoising step $T$, the hyper-parameter space (design space) of Algorithm 1 is defined as

$$\mathcal{H}_{\text{TFG}} = \{(N_{\text{recur}}, N_{\text{iter}}, \bar{\gamma}, \boldsymbol{\rho}, \boldsymbol{\mu}) : N_{\text{recur}}, N_{\text{recur}} \in \mathbb{N}, \bar{\gamma} \geq 0, \boldsymbol{\rho}, \boldsymbol{\mu} \in (\mathbb{R}_+ \cup \{0\})^T\}. \tag{6}$$

We use $\mathcal{H}_{\text{TFG}}$ to represent the complete hyper-parameter space and $\mathcal{H}_{\text{TFG}}(N_{\text{recur}} = N_0)$ to represent the subspace constrained on $N_{\text{recur}} = N_0$.

Definition 3.1 defines the hyper-parameter space spanned by TFG, where one hyper-parameter in $\mathcal{H}_{\text{TFG}}$ is an instantiation of the framework. Intuitively, $N_{recur}$ controls the recurrence of the algorithm, $N_{iter}$ controls the iterating when computing $\Delta_0$ (Line 8), $\bar{\gamma}$ controls the extent we smooth the original guidance function $f$ (Line 4), and $\boldsymbol{\rho}, \boldsymbol{\mu}$ control the strength of two types of guidance (Lines 7 and 8). A comprehensive explanation of the effect of each hyper-parameter can be found in Section 3.2.

Below is the major theorem showing that all algorithms presented in Section 2.1 correspond to special cases of TFG, thus unifying them into our framework and obviating the need for separate analyses.

**Theorem 3.2.** *The hyper-parameter space of*

- *MPGD [18] $\mathcal{H}_{MPGD}$ is equivalent to $\mathcal{H}_{TFG}(N_{recur} = N_{iter} = 1, \boldsymbol{\rho} = \boldsymbol{0}, \bar{\gamma} = 0)$.*
- *LGD [63] $\mathcal{H}_{LGD}$ is equivalent to $\mathcal{H}_{TFG}(N_{recur} = 1, N_{iter} = 0, \boldsymbol{\mu} = \boldsymbol{0})$.*

- *UGD [2] $\mathcal{H}_{UGD}$ is equivalent to $\mathcal{H}_{TFG}(\bar{\gamma} = 0)$.*
- *DPS [6] $\mathcal{H}_{DPS}$ is equivalent to $\mathcal{H}_{TFG}(N_{recur} = 1, N_{iter} = 0, \boldsymbol{\mu} = \mathbf{0}, \bar{\gamma} = 0)$.*
- *FreeDoM [78] $\mathcal{H}_{FreeDoM}$ is equivalent to $\mathcal{H}_{TFG}(N_{iter} = 0, \boldsymbol{\mu} = \mathbf{0}, \bar{\gamma} = 0)$.*

The complete analysis and proof of Theorem 3.2 is postponed to Appendix C. It implies that existing algorithms are limited in expressivity, covering only a subset of $\mathcal{H}_{TFG}$. In contrast, TFG covers the entire space and is guaranteed to perform better. In addition, TFG streamlines nuances between existing methods, allowing for a unified way to compare and study different techniques. Consequently, the versatile framework that TFG provides can simplify its adaptation to various applications.

## 3.2 Algorithm and design space analysis

We now present a concrete analysis of TFG and its design space $\mathcal{H}$ in detail. Similar to standard classifier-based guidance, TFG guides $\boldsymbol{x}_t$ at each denoising step $t$. To provide appropriate and informative guidance, TFG essentially leverages four techniques for guidance: Mean Guidance (Line 8) controlled by $N_{\text{iter}}, \boldsymbol{\mu}$, Variance Guidance (Line 7) controlled by $\boldsymbol{\rho}$, Implicit Dynamic (Line 4) controlled by $\bar{\gamma}$, and Recurrence (Line 5) controlled by $N_{\text{recur}}$.

**Mean Guidance** computes the gradient of $\tilde{f}(\boldsymbol{x})$ to $\boldsymbol{x}_{0|t}$ and is the most straightforward approach. However, this method can yield inaccurate guidance. To show this, notice that under perfect optimization we have $\boldsymbol{x}_{0|t} = \mathbb{E}[\boldsymbol{x}_0|\boldsymbol{x}_t]$, and when $p_0(\mathbb{E}[\boldsymbol{x}_0|\boldsymbol{x}_t])$ is close to zero, the predictor has rarely been trained on data from the region close to $\boldsymbol{x}_{0|t}$, making the gradient unstable and noisy. To mitigate this, one can iteratively add gradients of $\tilde{f}(\boldsymbol{x})$ to $\boldsymbol{x}_{0|t}$, encouraging $\boldsymbol{x}_{0|t}$ to escape low-probability regions.

**Variance Guidance** provides an alternative approach for improving the gradient estimation. The reason why we dub it variance guidance might be ambiguous, as the only difference is that the gradient is taken with respect to $\boldsymbol{x}_t$ (Line 7) instead of $\boldsymbol{x}_{0|t}$ (Line 8). The lemma below demonstrates that this essentially corresponds to a covariance re-scaled guidance.

**Lemma 3.3.** *If the model is optimized perfectly, i.e., $\epsilon_\theta(\boldsymbol{x}, t) = -\sqrt{1 - \bar{\alpha}_t}\nabla \log p_t(\boldsymbol{x})$, we have*

$$\Delta_t = \frac{\sqrt{\bar{\alpha}_t}}{1 - \alpha_t}\boldsymbol{\Sigma}_{0|t}\nabla_{\boldsymbol{x}_{0|t}}\tilde{f}(\boldsymbol{x}_{0|t}), \tag{7}$$

*where $\boldsymbol{\Sigma}_{0|t} \triangleq \int_{\boldsymbol{x}} p_{0|t}(\boldsymbol{x}|\boldsymbol{x}_t)(\boldsymbol{x} - \mathbb{E}[\boldsymbol{x}_0|\boldsymbol{x}_t])(\boldsymbol{x} - \mathbb{E}[\boldsymbol{x}_0|\boldsymbol{x}_t])^\top d\boldsymbol{x}$ is the covariance of $\boldsymbol{x}_0|\boldsymbol{x}_t$.*

Lemma 3.3 suggests that variance guidance refines mean guidance by incorporating the second-order information of $\boldsymbol{x}_0|\boldsymbol{x}_t$, specifically considering the correlation among components within $\boldsymbol{x}_{0|t}$. Consequently, positively correlated components could have guidance mutually reinforced, while negatively correlated components could have guidance canceled. This also implies that mean guidance and variance guidance are intrinsically leveraging different orders of information for guidance. In TFG, variance guidance is controlled by $\boldsymbol{\rho}_t$.

**Implicit Dynamic** transforms the predictor $f$ into its convolution via a Gaussian kernel $\mathcal{N}(\mathbf{0}, \bar{\gamma}(1 - \bar{\alpha}_t)\boldsymbol{I})$. This operation is initially introduced by LGD [63] to estimate $p_{0|t}(\boldsymbol{x}_0|\boldsymbol{x}_t)$. However, it is unclear why the form is preselected as a Gaussian distribution. We argue that this technique is effective because it creates an implicit dynamic on $\boldsymbol{x}_{0|t}$. Specifically, starting from $\boldsymbol{x}_{0|t}$, it iteratively adds noise to $\boldsymbol{x}_{0|t}$, evaluates gradient, and moves $\boldsymbol{x}_{0|t}$ based on the gradient. The repeating process converges to the density proportional to $f(\boldsymbol{x})$ when $N_{\text{iter}}$ goes to infinity, driving $\boldsymbol{x}_{0|t}$ to high-density regions. This explanation is justified by Table 1: the performance remains nearly un-

Table 1: Influence of the number of Monte-Carlo samples in estimating the expectation of Line 4. Both the FID and the accuracy remain unchanged when #Samples varies, suggesting that the number of samples is less important. More details are in Appendix E.1.

| #Samples | Variance only | | Mean only | |
|---|---|---|---|---|
| | FID | Acc(%) | FID | Acc(%) |
| 1 | 90.6 | 65.8 | 101 | 36.2 |
| 2 | 91.0 | 65.2 | 100 | 35.6 |
| 4 | 90.7 | 64.9 | 99.7 | 36.2 |

changed as we gradually decrease the number of Monte-Carlo samples in estimating the expectation (Line 4) down to 1, implying that the *preciseness* of estimation is not essential, but adding noises is.

**Recurrence** helps strengthen the guidance by iterating the previous three techniques to obtain $\boldsymbol{x}_{t-1}$ and resample $\boldsymbol{x}_t$ back and forth. This can be understood as an *Ornstein–Uhlenbeck process*[42] on $\boldsymbol{x}_{t-1}$ where Line 6∼9 corresponds to the drift term and $\boldsymbol{x}_{t-1} \rightarrow \boldsymbol{x}_t$ (Line 10) the white noise term. Intuitively, it finds a trade-off between the error inherited from previous steps (the more you recur, the

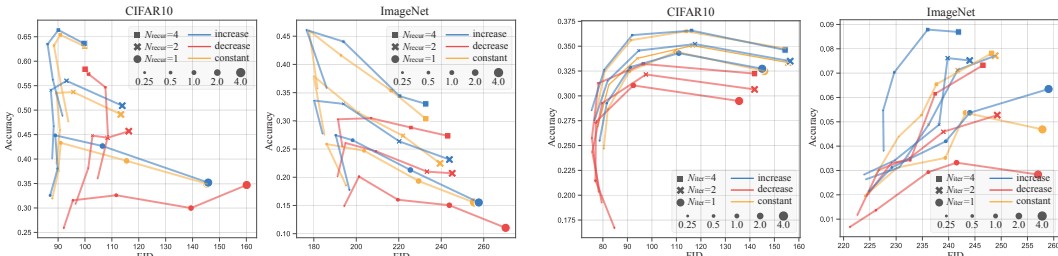

Figure 2: Comparison of three structures in Equation (8) of $\boldsymbol{\rho}$ and $\boldsymbol{\mu}$ on CIFAR10 and ImageNet, under different choices of the rest hyper-parameters in $\mathcal{H}_{\text{TFG}}$. We set $\boldsymbol{\rho} = \mathbf{0}, \bar{\gamma} = 0$ when studying structures of $\boldsymbol{\mu}$, and similarly for $\boldsymbol{\rho}$. Results are averaged across all labels. The comparative relationship between structures remains unchanged when the rest of the parameters vary.

less previous error stays) and the accumulated error in this step (the more you recur, the more error in the current guidance you suffer). Empirically, we also find that the generation quality improves and then deteriorates as we increase $N_{\text{recur}}$.

# 4 Design Space of TFG: Analysis and Searching Strategy

Admittedly, a more extensive design space only yields a better performance if an effective and robust hyper-parameter searching strategy can be applied. For example, arbitrarily complex neural networks are guaranteed to have better optimal performance than simple linear models, but finding the correct model parameters is significantly more difficult. This section dives into this core problem by comprehensively analyzing the hyper-parameter space structure of $\mathcal{H}_{\text{TFG}}$, and further proposing a general searching algorithm applicable for any general downstream tasks.

The hyper-parameters of $\mathcal{H}_{\text{TFG}}$ can be categorized into two parts: time-dependent vectors $\boldsymbol{\rho}, \boldsymbol{\mu}$, and time-independent sacalars $N_{\text{recur}}, N_{\text{iter}}, \bar{\gamma}$. While a grid search can potentially result in the best performance, performing such an extensive search in $\mathcal{H}_{\text{TFG}}$ is highly impractical, especially considering the vector parameters $\boldsymbol{\rho}, \boldsymbol{\mu}$. Fortunately, below we demonstrate that, if we decompose $\boldsymbol{\rho}$ into $\bar{\rho} \cdot s_\rho(t)$ (same for $\boldsymbol{\mu}$) where $\bar{\rho}$ is a scalar and $s_\rho(t)$ is a "structure" (a non-negative function) such that $\sum_t s_\rho(t) = T$, then some structures are consistently better than others regardless of the other hyper-parameters. This allows us to pre-locate an appropriate structure for the given task and efficiently optimize the rest of the scalar hyper-parameters. Our analysis is conducted on the label guidance task on CIFAR-10 [30] and ImageNet [55], with experimental settings identical to Section 3.

**Structure analysis.** Motivated by the default structure selected in UGD and LGD, we consider three structures for both $s_\rho(t)$ and $s_\mu(t)$ as

$$s(t) = \frac{\alpha_t}{\sum_{t=1}^{T} \alpha_t}(\text{increase}), \ \ s(t) = \frac{(1 - \alpha_t)}{\sum_{t=1}^{T}(1 - \alpha_t)}(\text{decrease}), \ \ s(t) = 1(\text{constant}). \tag{8}$$

These structures are selected to be qualitatively different, while each is justified to be reasonable under certain conditions [18, 78, 2]. We leave the study of more structures to future works. The rest of the parameters are grid-searched for the comprehensiveness of the analysis. For $s_\rho(t)$, we set $N_{\text{recur}} = \{1, 2, 4\}$ and $\bar{\rho} = \{0.25, 0.5, 1.0, 2.0, 4.0\}$; and for $s_\mu(t)$, we set $N_{\text{iter}} = \{1, 2, 4\}$ and $\bar{\mu} = \{0.25, 0.5, 1.0, 2.0, 4.0\}$. We run label guidance for each configuration and each of the ten labels on CIFAR10 (four labels on ImageNet, due to computation constraints).

As presented in Figure 2, the relationship between different structures remains unchanged when the rest of the parameters vary. For instance, on both datasets, the Validity-FID performance curves consistently move top-left (implying a better performance) when we switch from "decrease" structure (red lines) to "constant" structure (yellow lines) to "increase" structure (blue lines) for both $\boldsymbol{\rho}, \boldsymbol{\mu}$ and different values of $N_{\text{recur}}$ and $N_{\text{iter}}$. This invariant relationship is essential as it allows for an efficient hyper-parameters search in $\mathcal{H}_{\text{TFG}}$ by first determining appropriate structures for $s_\rho(t), s_\mu(t)$ under a simple subspace, and then selecting the rest scalar parameters.

Table 2: List of 14 task types we benchmark. Each task is run with multiple individual targets (38 in total). We evaluate the guidance validity (how well a sample is aligned with the target predictor) and the guidance fidelity (how well a sample is aligned with the unconditional distribution) according to the task type.

| Diffusion Model | Task-ID | Targets | Guidance Validity | Guidance Fidelity |
|---|---|---|---|---|
| Cat-DDPM | Gaussian deblur | \ | LPIPS ↓ | FID ↓ |
| | Super-resolution | \ | LPIPS ↓ | FID ↓ |
| CelebA-DDPM | Combined guidance (gender+age) | 2 genders × 2 ages | Accuracy (%) ↑ | KID (log) ↓ |
| | Combined guidance (gender+hair) | 2 genders × 2 hair colors | Accuracy (%) ↑ | KID (log) ↓ |
| CIFAR10-DDPM | Label guidance (CIFAR10) | 10 labels $(0, \cdots, 9)$ | Accuracy (%) ↑ | FID ↓ |
| ImageNet-DDPM | Label guidance (ImageNet) | 4 labels $(111, \cdots, 444)$ | Accuracy (%) ↑ | FID ↓ |
| | Fine-grained guidance | 4 labels $(111, \cdots, 444)$ | Accuracy (%) ↑ | FID ↓ |
| Stable-Diffusion | Style transfer | 4 styles | Style score ↓ | CLIP score ↑ |
| Molecule-EDM | Quantum Properties (×6) | Property distribution | MAE ↓ | Valid ratio ↑ |
| Audio-Diffusion | Audio declipping | \ | DTW (%) ↓ | FAD ↓ |
| | Audio inpainting | \ | DTW (%) ↓ | FAD ↓ |

**Computation cost analysis.** Among the scalars parameters, $N_{\text{recur}}$ and $N_{\text{iter}}$ directly influence the total computational cost, while $\bar{\rho}, \bar{\mu}, \bar{\gamma}$ do not. With a certain range, performance increases when the value of $N_{\text{recur}}, N_{\text{iter}}$ increase[5], and the trade-off between generation quality and computation time is presented in Figure 3: recurrence leads to a $N_{\text{recur}}$ times cost with clear performance gain; iteration (on $x_{0|t}$) results in less increase of computation time, and its effect plateaus. In practice, users can determine their values based on computation resources, but an upper bound of 4 suffices to unlock a near-optimal performance.

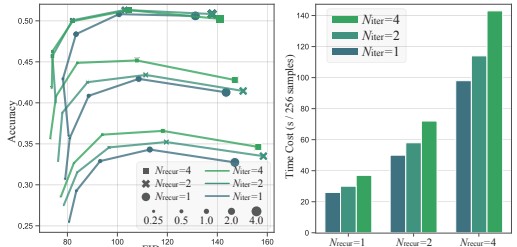

Figure 3: Accuracy and FID on CIFAR10 under different $N_{\text{recur}}$ and $N_{\text{iter}}$. $s_\rho(t), s_\mu(t)$ are fixed to "increase" structure, and $\boldsymbol{\rho} = \boldsymbol{\gamma} = 0$.

**Searching strategy.** The above analysis successfully simplify the task-specific hyper-parameter search problem without significant performance sacrifice. It remains to be decided the scalar values $\bar{\rho}, \bar{\mu}, \bar{\gamma}$. Here we propose a strategy based on beam search to effectively and efficiently select their values. Specifically, our searching strategy starts with an initial set $T = \{(\bar{\rho}_{\text{init}}, \bar{\mu}_{\text{init}}, \bar{\gamma}_{\text{init}})\}$, where these initial values are small enough to approximate TFG as an unconditional generation. At each searching step, for each tuple in $T$, we separately double the values of $\bar{\rho}, \bar{\mu},$ and $\bar{\gamma}$ to generate up to $3|T|$ new configurations. We conduct a small-sized generation trial for each new configuration and update $T$ to be the top $K$ configurations with the highest evaluation results that are determined by user requirements (e.g., accuracy, FID, or a combination). This iterative process is repeated until $T$ stabilizes or the maximum number of search steps is reached. Notice that this process is conducted with a much smaller sample size, and consequently, the computation time is highly controllable.

## 5 Benchmarking

This section comprehensively benchmarks training-free guidance under the TFG framework and the design searching strategy in Section 4. We consider 7 datasets, 16 different tasks, and 40 individual targets with a total experimental cost of more than 2,000 A100 GPU hours. For comparison, we also run experiments for each of the existing methods (where the design searching is conducted in the corresponding subspace). All methods, tasks, search strategies, and evaluations are unified in our codebase, with details specified in Appendices D and E.

### 5.1 Settings

**Diffusion models.** (1) **CIFAR10-DDPM** [48] is a U-Net [54] model trained on CIFAR10 [30] images. (2) **ImageNet-DDPM** [7] is an larger U-Net model trained on ImageNet-1k [55] images. (3) **Cat-DDPM** is trained on Cat [12] images. (4) **CelebA-DDPM** is trained on CelebA-HQ dataset [26] that consists millions of human facial images. (5) **Molecule-EDM** [24] is an equivariant diffusion

---

[5]We notice that the FID worsens when $N_{\text{recur}}$ or $N_{\text{iter}}$ are too large (e.g., 10).

Table 3: Benchmarking TFG and existing algorithms on 16 task types and 40 individual targets. Each cell presents the *guidance validity/generation fidelity* averaged across multiple targets in the task (*e.g.*, labels, image styles). The best guidance validity is **bold**, and the second best underline. The relative improvement of guidance validity is computed between TFG and the existing method with the highest guidance validity.

| Task-ID | DPS | LGD | FreeDoM | MPGD | UGD | TFG | Rel. Improvement |
|---|---|---|---|---|---|---|---|
| Deblur $(\downarrow,\downarrow)$ | 0.390 / 98.3 | 0.270 / 85.1 | 0.245 / 87.4 | 0.177 / 69.3 | 0.200 / 69.3 | **0.150** / 64.5 | +15.3% |
| Super resolution $(\downarrow,\downarrow)$ | 0.420 / 109 | 0.360 / 96.7 | 0.191 / 74.5 | 0.283 / 82.0 | 0.249 / 75.9 | **0.190** / 65.9 | +0.524% |
| Gender+Age $(\uparrow,\downarrow)$ | 71.6 / -4.26 | 52.0 / -5.10 | 68.7 / -3.89 | 68.6 / -4.79 | 75.1 / -4.37 | **75.2** / -3.86 | +0.133% |
| Gender+Hair $(\uparrow,\downarrow)$ | 73.0 / -3.90 | 55.0 / -5.00 | 67.1 / -3.50 | 63.9 / -4.33 | 71.3 / -4.12 | **76.0** / -3.60 | +4.11% |
| CIFAR10 $(\uparrow,\downarrow)$ | 50.1 / 172 | 32.2 / 102 | 34.8 / 135 | 38.0 / 88.3 | 45.9 / 94.2 | **52.0** / 91.7 | +3.59% |
| ImageNet $(\uparrow,\downarrow)$ | 38.8 / 193 | 11.5 / 210 | 19.7 / 200 | 6.80 / 239 | 25.5 / 205 | **40.9** / 176 | +5.41% |
| Fine-grained $(\uparrow,\downarrow)$ | 0.00 / 348 | 0.48 / 246 | 0.58 / 258 | 0.58 / 249 | 1.07 / 255 | **1.27** / 256 | +18.7% |
| Style Transfer $(\downarrow,\uparrow)$ | 5.06 / 31.7 | 5.42 / 31.3 | 5.26 / 31.2 | 4.08 / 31.5 | 4.97 / 31.5 | **3.16** / 29.0 | +22.5% |
| Polarizability $\alpha$ $(\downarrow,\uparrow)$ | 51169.7 / 92.3 | 7.155 / 84.3 | 5.922 / 88.0 | 4.26 / 88.4 | 5.45 / 73.8 | **3.90** / 84.2 | +8.45% |
| Dipole $\mu$ $(\downarrow,\uparrow)$ | 63.2 / 77.3 | 1.51 / 86.6 | 1.35 / 89.5 | 1.51 / 73.5 | 1.56 / 57.6 | **1.33** / 74.9 | +1.48% |
| Heat capacity $C_v$ $(\downarrow,\uparrow)$ | 5.26 / 78.4 | 3.77 / 77.1 | 2.84 / 90.9 | 2.86 / 86.1 | 3.02 / 84.0 | **2.77** / 85.5 | +2.57% |
| Highest MO energy $\epsilon_{\text{HOMO}}$ $(\downarrow,\uparrow)$ | 0.744 / 83.8 | 0.664 / 66.4 | 0.623 / 62.3 | **0.554** / 53.4 | 0.582 / 58.2 | 0.568 / 77.3 | -2.53% |
| Lowest MO energy $\epsilon_{\text{LUMO}}$ $(\downarrow,\uparrow)$ | NA / NA | 1.20 / 90.9 | 1.16 / 90.2 | 1.06 / 82.2 | 1.27 /85.1 | **0.984** / 80.1 | +7.17% |
| MO energy gap $\epsilon_\Delta$ $(\downarrow,\uparrow)$ | 1.38 / 75.7 | 1.19 / 85.3 | 1.17 / 88.5 | 1.07 / 72.5 | 1.15 / 75.7 | **0.893** / 62.5 | +16.7% |
| Audio declipping $(\downarrow,\downarrow)$ | 633 / 3.60 | 157 / 2.33 | 126 / 0.173 | 178 / 0.402 | 150 / 0.262 | **101** / 0.172 | +19.8% |
| Audio inpainting $(\downarrow,\downarrow)$ | 643 / 4.71 | 103 / 2.22 | 41.3 / 0.08 | 608 / 4.63 | 116 / 0.53 | **36.3** / 0.06 | +12.1% |

model pretrained on molecule dataset QM9 [50] that performs molecule generation from scratch. (6) **Stable-Diffusion** (*v1.5*) [53] is a latent text-to-image model that generate images with text prompts. (7) **Audio-Diffusion**[6] is a audio diffusion model based on DDPM trained to generate mel spectrograms of 256x256 corresponding to 5 seconds of audio.

**Tasks.** Our tasks (Table 2) cover a wide range of interests, including Gaussian deblur, super-resolution, label guidance, style transfer, molecule property guidance, audio declipping, audio inpainting, and guidance combination. Each task is run on multiple datasets or with multiple targets (*e.g.*, different labels, molecular properties, styles).

**Other settings.** We consistently set the time step $T = 100$ and the DDIM parameter $\eta = 1$. We consider $N_{\text{recur}} = 1, N_{\text{iter}} = 4$ and use a single sample for Implicit Dynamic (Line 4) throughout all experiments and methods for fair comparison. For TFG, the structures of $\rho$ and $\mu$ are set to "increase" and the scalars $\bar{\rho}, \bar{\mu}, \bar{\gamma}$ are determined via our searching strategy. We follow the setting in original papers if they specify their hyper-parameters. blueFor specific tricks in the code that are not mentioned in papers, we choose to align with original papers. Otherwise, values are determined via searching with $1/8$ of the sample size and a maximum search step of 6. For fairness of comparison, we use accuracy as the metric during the search and compare different algorithms on the metric, but we report both accuracy and FID.

## 5.2 Benchmarking results

We compare all six methods in Table 3. TFG outperforms existing algorithms in 13 over 14 settings, achieving an average guidance validity improvement of 7.4% compared to the best existing algorithm. Notice that we do not compare with the best algorithm in terms of generation fidelity because obtaining high realness samples is not our objective in training-free guidance, and an unconditional model suffices to generate high realness samples (with extremely low validity). Interestingly, different methods achieve the second best performance on different tasks, suggesting the variance of these methods, while TFG is consistent thanks to the unification.

We want to highlight that despite the superior performance of TFG, the key intention of our experiments is not restrained to comparing TFG with existing methods, but more importantly to systematically benchmark under the training-free guidance setting to see how much we have achieved in various tasks with different difficulties. Below we go through each task separately and conduct relevant ablation studies to provide a more fine-grained analysis.

**Fine-grained label guidance.** In addition to the standard label guidance, we for the first time study the *out-of-distribution* fine-grained label guidance under the training-free setting, a problem where no existing training-based methods are available. We consider the bird-species guidance using an

---
[6]https://huggingface.co/teticio/audio-diffusion-256

EfficienNet trained to classify 525 fine-grained bird species. This problem remains highly difficult for leading text-to-image generative models such as DALLE. Under recurrence, TFG can generate at most 2.24% of accurate birds, compared with the unconditional generation rate of 0.

**Recurrence on label guidance.** We go back to the failure case we study in Section 3, *i.e.*, the standard label guidance problem on CIFAR10 where the training-based method offers an 85% accuracy, while the accuracy of TFG without recurrence accuracy is 52% only. As presented in Table 4, increasing $N_{\text{recur}}$ significantly closes the gap from 33% to 8%. Similar improvement is observed in other datasets as well.

Table 4: The accuracy / FID for TFG with different recurrence step $N_{\text{recur}}$ on three label guidance datasets, averaged across all labels.

| Recurrence | 1 | 2 | 4 |
|---|---|---|---|
| CIFAR10 | 52.0 / 91.7 | 66.8 / 88.7 | 77.1 / 73.9 |
| ImageNet | 40.9 / 177 | 52.3 / 163 | 59.8 / 165 |
| Fine-grained | 1.27 / 256 | 1.66 / 259 | 2.24 / 259 |

**Multiple guidance and bias mitigation.** We next consider the scenario with multiple targets: control the generation of human faces based on gender and hair color (or age) using two predictors. It is well known that the label imbalance in CelebA-HQ causes classifiers to focus on *spurious correlations [76]*, such as using hair colors to classify gender, a biased feature we aim to avoid. The stratified performance of TFG on "gender + age" and "gender + hair" guidance are presented in Table 5. Despite the highly disparate performance, training-free guidance largely alleviates the imbalance: only 1% of images in CelebA are "male + blonde hair", while the generated accuracy is 46.7%.

Table 5: The accuracy of multi-label guidance on CelebA, where labels 0 and 1 correspond to female and male (gender), non-blonde and blonde (hair color), and young and old (age). The accuracy is lower for minority groups, indicating an implicit bias in the generation process. Despite this, it is still much higher than unconditional generation.

| Target label | 0+0 | 0+1 | 1+0 | 1+1 |
|---|---|---|---|---|
| gender + hair | 92.2 | 72.7 | 89.8 | 46.7 |
| gender + age | 92.9 | 73.6 | 93.6 | 69.1 |

**Molecule property guidance.** To our knowledge, we are the first to study training-free guidance for molecule generation. We interestingly find in Table 3 that TFG is effective in guiding molecules towards desirable properties, yielding the highest guidance validity on 5 out of 6 targets with 5.64% MAE improvement over existing methods, verifying the generality of our approach as a unified framework in completely unseen domains. Notice that, unlike images, molecules with better validity usually have lower generation fidelity, a finding reflected in previous work [3].

**Audio Guidance.** We extend our investigation to the audio modality, where TFG achieves significant relative improvements over existing methods. Given that the audio domain is rarely explored in training-free guidance literature, our benchmarks will contribute to future research in this area.

# 6 Discussions and Limitations

Recently, training-free guidance for diffusion models has gained increasing attention and has been adopted in various applications. TFG is based on an extensive literature review over ten algorithmic papers for different purposes, including images, audio, molecules, and motions [34, 8, 43, 32, 19, 13, 16, 15, 45, 38, 68]. While we incorporate several key algorithms into our framework, we acknowledge that encompassing all approaches is impossible, as it would make the unification bloated and less practical. We seek to find a balance point by unifying most representative algorithms while keeping the techniques clear and easily studied.

An often discussed problem is why we care about training-free guidance, given the ever-growing community of language-based generative models such as the image generator of GPT4. In practice, there are countless conditional generation tasks where the conditions are hard to accurately convey to or represent by language encoders. For instance, it can fail to under a complex property of a molecule or generate CelebA-style faces. We give an illustrative analysis in Appendix A.1. Despite that training-free guidance is important, this paper does not systematically analyze what types of conditional generation are, in general, more suitable for the framework and what types are for language-based models. That said, training-free guidance is fundamentally difficult due to the misalignment between the training objective of target predictors and the diffusion, with a more detailed discussion in Appendix A.2. This paper does not comprehensively analyze this misalignment, and the gap between training-based and TFG remains high in some tasks like molecule property guidance. We hope that future works can analytically dive into these problems.

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

# A   The Motivation of Studying Training-free Guidance

In this section, we argue that training-free guidance using off-the-shelf models is a crucial and timely research problem deserving more attention and effort. We begin by providing an illustrative analysis that highlights the limitations of current strong text-to-image generative models, underscoring the necessity for training-free guidance. Furthermore, we assert that training-free guidance remains a significant challenge, with previous literature underestimating its complexity. Given its necessity and inherent difficulties, we call for increased focus from the research community on this problem and offer our benchmarks and code base to help accelerate progress in this area.

## A.1   Failure case of image generation with GPT4

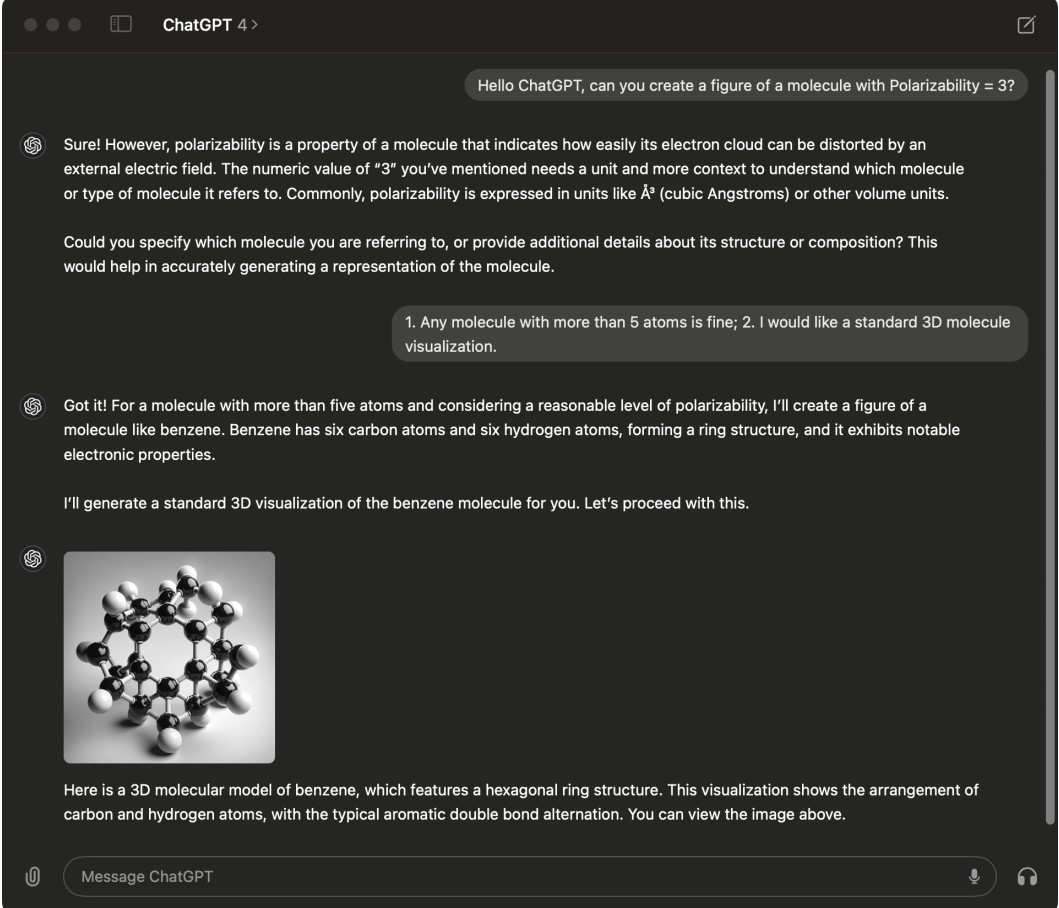

Figure 4: **Prompting GPT4 to generate property guided molecules**. It is hard for the image generator to understand the target and generate faithful samples. In this dialog, GPT4 claims to generate a benzene molecule but the sample is apparently not a benzene. There are also many invalid carbon atoms with more than 4 bonds and the polarizability target is not achieved.

**It's hard for GPT4 to understand targets.**   In Figure 4, we ask GPT-4 to generate a molecule with polarizability $\alpha = 3$, which is a task we use to evaluate training-free guidance (refer to Figure 16 for visualization). We found that the GPT-4 generated molecule is apparently invalid and unrealistic: the generated molecule contains many carbon atoms with more than 4 bonds (the maximum allowed number is 4); and the generated molecule is apparently not a benzene which is claimed by the text outputs. From this case we may understand that it is hard to follow diverse user-defined instructions for the foundational generative models, where the user-defined targets may be subtle, fine-grained, combinatorial, and open-ended.

To this end, training-free guidance offers two key advantages: (1) It allows for greater specificity in target requirements by enabling the use of a differentiable objective function, making the generation process more steerable; (2) The objective function is plug-and-play, facilitating the addition of new targets and tasks to a pre-trained generative model. Since there is no need to retrain the diffusion or prediction models, this approach makes the generative process lightweight and applicable to various downstream tasks.

**It's hard for GPT4 to capture the targeted distribution.**     Another important metrit for training-free guidance is the flexibility of choosing diffusion models. For the same target, we can switch from different diffusion models to change the unconditional background distribution. For example, it is hard for GPT4 to generate CelebA-like samples though it "knows CelebA dataset very well":

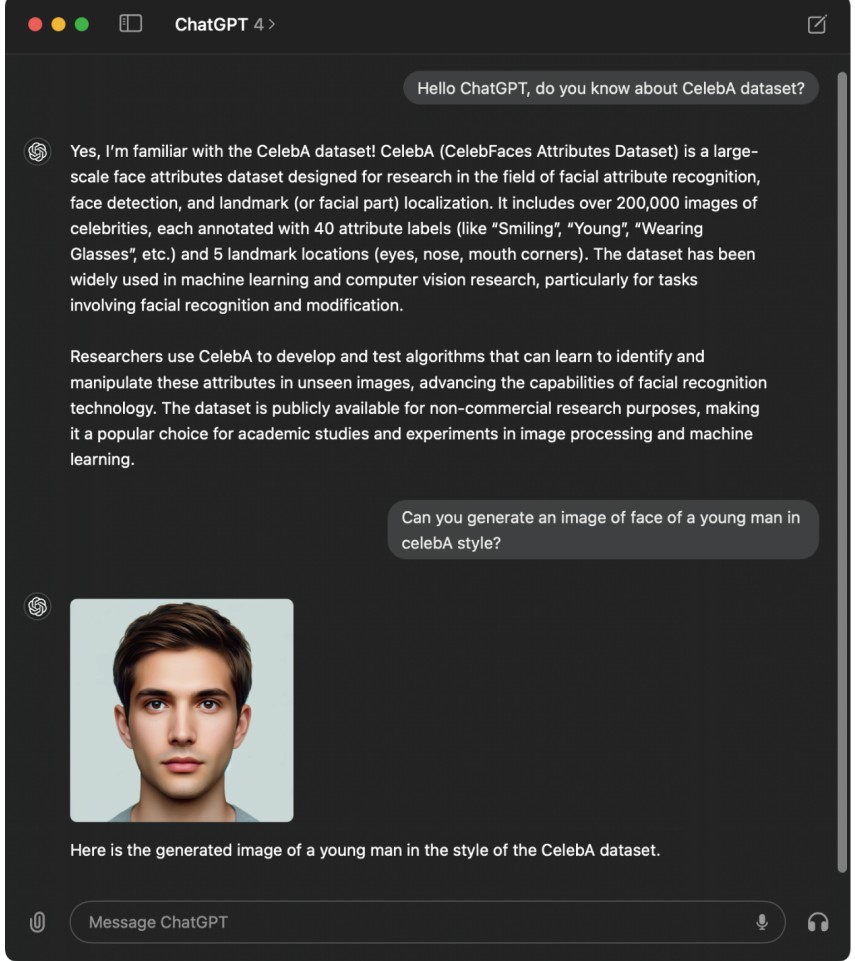

Figure 5: **Prompting GPT4 to generate CelebA-like images.** We first prompt ChatGPT to probe its knowledge of CelebA dataset and then ask it to generate a young man figure in CelebA style. However, the generated figure is apprently not in the distribution of CelebA (refer to Figure 12) for comparison.

The flexibility to use different diffusion models provides an opportunity to generate a wider range of user-defined targets. With training-free guidance, individuals can select their preferred diffusion model to establish the background distribution and use the prediction model to steer the generation towards specific properties. This approach may represent a future direction for human-AI interaction.

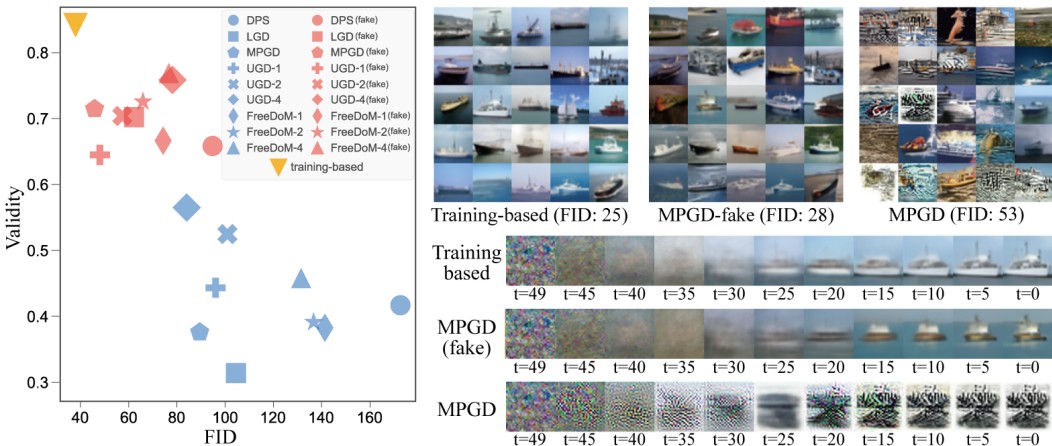

Figure 6: (Left) The accuracy and FID of different methods under different settings on CIFAR10 [30], average across ten labels and 2048 samples per label. The suffix number in UGD and FreeDoM represents recurrent step $N_{recur}$, and (fake) stands for a synthetic setting where we apply a training-based classifier but set $t = 0$ and use the same training-free guidance methods. A huge performance gap between different settings suggests the intrinsic difficulty of training-free guidance. (Right) Illustration of generated "ship" using MPGD under different settings (top) and the sampling trajectory of the predicted clean image $x_{0|t}$ (down).

## A.2 The fundamental challenge of training-free guidance

Despite the array of algorithms available and their reported successes in various applications, we conduct a case study on CIFAR10 [30] to illustrate the challenging nature of training-free guidance and the insufficiency of existing methods. Specifically, we compare the training-based approach and training-free approach for the label guidance task on CIFAR10, with the diffusion model pretrained by [7], the training-based time-dependent classifier $f(x, t)$ by [7], and the training-free standard label classifier $f(x)$ pretrained only on *clean* CIFAR10 by [9]. and a "fake" training-free classifier defined as $f(x, t)|_{t=0}$. The first serves as the oracle benchmark, while the second corresponds to the standard training-free guidance. The third setting, as considered in LGD [63], uses a "fake" training-free classifier since its parameters are shared across different time steps $t$ during training, resulting in an implicit regularization that is not available for practical predictors. This setting serves as a comparison to help identify the difficulty of training-free guidance.

Quantitative and qualitative results are shown in Figure 6. All training-free approaches significantly under-perform training-based guidance, with a significant portion of generated images being highly unnatural (when guidance is strong) or irrelevant to the label (when guidance is weak). A more clear illustration on this can be found in Figure 7. In terms of "fake" classifiers, it leads to a remarkable difference from real training-free classifiers even under identical experimental settings. It generates

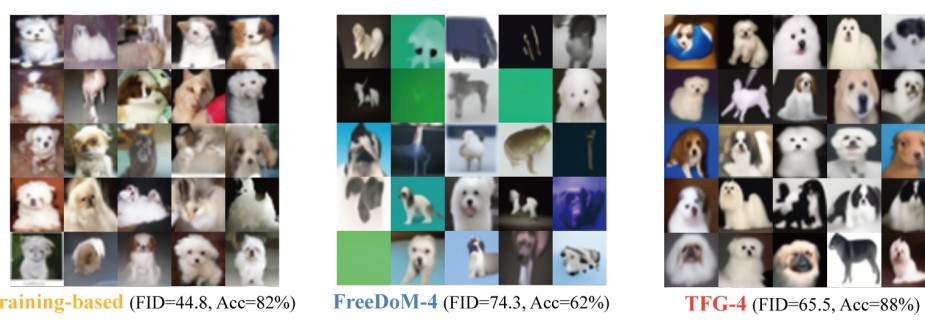

Training-based (FID=44.8, Acc=82%)    FreeDoM-4 (FID=74.3, Acc=62%)    TFG-4 (FID=65.5, Acc=88%)

Figure 7: Illustration on CIFAR10 dogs generated with different algorithms. Compared with training-based method, training-free methods fall behind but TFG significantly outperforms existing methods.

much less messy images due to the implicit regularization from the training process, where noisy images are also "seen" (although $t$ is fixed to $0$ upon guidance). Unfortunately, such types of predictors are inaccessible in practice (otherwise, we can use classifier-based guidance directly). From the comparison, we know that the key challenge of training-free guidance is the lack of a "smoothing" classifier that can produce faithful guidance in the "unseen" noisy image space.

The observation largely uncovers the essential difficulties of training-free guidance, and motivates us to systematically study techniques that can improve generation quality. Unfortunately, comparisons between existing techniques are ambiguous since different methods are tested on distinct and primarily qualitative applications, which in turn hinders the in-depth study in this field. To this end, we resolve to revisit this complicated scenario and design a clear and comprehensive framework for training-free guidance.

# B Pseudo-code and schematics

We have presented the pseudo-code of TFG in Algorithm 1. Below, we provide a copy of the DPS (Algorithm 2), MPGD (Algorithm 3), FreeDoM (Algorithm 4), UGD (Algorithm 5), and LGD (Algorithm 6). Notice that LGD does not provide a pseudo-code, and we present their algorithm following their paper as a modification of DPS. We do not change the original algorithms' notations for reference. Please see the proof in Appendix C for the equivalence analysis. We provide a schematic of existing algorithms in Figure 8.

---

**Algorithm 2** DPS - Gaussian

---

**Require:** $N, y, \{\zeta_i\}_{i=1}^{N}, \{\tilde{\sigma}_i\}_{i=1}^{N}$
  $x_N \sim \mathcal{N}(0, I)$
  **for** $i = N - 1$ **to** $0$ **do**
    $\hat{s} \leftarrow s_\theta(x_i, i)$
    $\hat{x}_0 \leftarrow \frac{1}{\sqrt{\bar{\alpha}_i}}(x_i + (1 - \bar{\alpha}_i)\hat{s})$
    $z \sim \mathcal{N}(0, I)$
    $x'_{i-1} \leftarrow \frac{\sqrt{\alpha_i}(1-\bar{\alpha}_{i-1})}{1-\bar{\alpha}_i}x_i + \frac{\sqrt{\bar{\alpha}_{i-1}}\beta_i}{1-\bar{\alpha}_i}\hat{x}_0 + \tilde{\sigma}_i z$
    $x_{i-1} \leftarrow x'_{i-1} - \zeta_i \nabla_{x_i} \|y - \mathcal{A}(\hat{x}_0)\|_2^2$
  **end for**
  **return** $\hat{x}_0$

---

**Algorithm 3** MPGD for pixel diffusion models

---

1:  $x_T \sim \mathcal{N}(0, I)$
2:  **for** $t = T, \dots, 1$ **do**
3:    $\epsilon_t \sim \mathcal{N}(0, I)$
4:    $x_{0|t} = \frac{1}{\sqrt{\bar{\alpha}_t}}(x_t - \sqrt{1 - \bar{\alpha}_t}\epsilon_\theta(x_t, t))$
5:    **if** requires manifold projection **then**
6:       $x_{0|t} = g_{\mathcal{M}}(x_{0|t}, L(x_{0|t}; y), c_t)$
7:    **else**
8:       $x_{0|t} = x_{0|t} - c_t \nabla_{x_{0|t}} L(x_{0|t}; y)$
9:    **end if**
10:   $x_{t-1} = \sqrt{\bar{\alpha}_{t-1}}x_{0|t}$
11:     $+\sqrt{1 - \bar{\alpha}_{t-1} - \sigma_t^2}\epsilon_\theta(x_t, t) + \sigma_t\epsilon_t$
12: **end for**
13: **return** $x_0$

---

**Algorithm 4** FreeDoM + Efficient Time-Travel Strategy

---

**Require:** condition $\mathbf{c}$, unconditional score estimator $s(\cdot, t)$, time-independent distance measuring function $\mathcal{D}_{\boldsymbol{\theta}}(\mathbf{c}, \cdot)$, pre-defined parameters $\beta_t, \bar{\alpha}_t$, learning rate $\rho_t$, and the repeat times of time travel of each step $\{r_1, \cdots, r_T\}$.
  $\mathbf{x}_T \sim \mathcal{N}(\mathbf{0}, \mathbf{I})$
  **for** $t = T, \dots, 1$ **do**
    **for** $i = r_t, \dots, 1$ **do**
      $\boldsymbol{\epsilon}_1 \sim \mathcal{N}(\mathbf{0}, \mathbf{I})$ if $t > 1$, else $\boldsymbol{\epsilon}_1 = \mathbf{0}$.
      $\mathbf{x}_{t-1} = (1 + \frac{1}{2}\beta_t)\mathbf{x}_t + \beta_t s(\mathbf{x}_t, t) + \sqrt{\beta_t}\boldsymbol{\epsilon}_1$
      $\mathbf{x}_{0|t} = \frac{1}{\sqrt{\bar{\alpha}_t}}(\mathbf{x}_t + (1 - \bar{\alpha}_t)s(\mathbf{x}_t, t))$
      $\boldsymbol{g}_t = \nabla_{\mathbf{x}_t}\mathcal{D}_{\boldsymbol{\theta}}(\mathbf{c}, \mathbf{x}_{0|t}(\mathbf{x}_t)))$
      $\mathbf{x}_{t-1} = \mathbf{x}_{t-1} - \rho_t \boldsymbol{g}_t$
      **if** $i > 1$ **then**
        $\boldsymbol{\epsilon}_2 \sim \mathcal{N}(\mathbf{0}, \mathbf{I})$
        $\mathbf{x}_t = \sqrt{1 - \beta_t}\mathbf{x}_{t-1} + \sqrt{\beta_t}\boldsymbol{\epsilon}_2$
      **end if**
    **end for**
  **end for**
  **return** $\mathbf{x}_0$

---

---

**Algorithm 5** Universal Guidance (its $\alpha$ is our $\bar{\alpha}$)

---

**Parameter:** Recurrent steps $k$, gradient steps $m$ for backward guidance and guidance strength $s(t)$,
**Required:** $z_T$ sampled from $\mathcal{N}(0, I)$, diffusion model $\epsilon_\theta$, noise scales $\{\alpha_t\}_{t=1}^T$, guidance function $\mathbf{f}$, loss function $\ell$, and prompt $c$
**for** $t = T, T-1, \ldots, 1$ **do**
    **for** $n = 1, 2, \ldots, k$ **do**
        Calculate $\hat{z}_0$ as $\frac{z_t - (\sqrt{1-\alpha_t})\epsilon_\theta(z_t, t)}{\sqrt{\alpha_t}}$
        $\hat{\epsilon}_\theta(z_t, t) = \epsilon_\theta(z_t, t) + s(t) \cdot \nabla_{z_t} \ell(c, \mathbf{f}(\hat{z}_0))$
        **if** $m > 0$ **then**
            Calculate $\Delta z_0$ by minimizing $\ell(c, \mathbf{f}(\hat{z}_0 + \Delta))$. with $m$ steps of gradient descent
            Perform backward universal guidance by $\hat{\epsilon}_\theta \leftarrow \hat{\epsilon}_\theta - \sqrt{\alpha_t/(1-\alpha_t)}\Delta z_0$
        **end if**
        $z_{t-1} \leftarrow S(z_t, \hat{\epsilon}_\theta, t)$
        $\epsilon' \sim \mathcal{N}(0, I)$
        $z_t \leftarrow \sqrt{\alpha_t/\alpha_{t-1}} z_{t-1} + \sqrt{1 - \alpha_t/\alpha_{t-1}}\epsilon'$
    **end for**
**end for**

---

**Algorithm 6** LGD (from DPS)

---

**Require:** $N, y, \{\zeta_i\}_{i=1}^N, \{\tilde{\sigma}_i\}_{i=1}^N, n$
  $x_N \sim \mathcal{N}(0, I)$
  **for** $i = N - 1$ **to** $0$ **do**
    $\hat{s} \leftarrow s_\theta(x_i, i)$
    $\hat{x}_0 \leftarrow \frac{1}{\sqrt{\bar{\alpha}_i}}(x_i + (1 - \bar{\alpha}_i)\hat{s})$
    $z \sim \mathcal{N}(0, I)$
    $x_{i-1} \leftarrow \frac{\sqrt{\alpha_i}(1-\bar{\alpha}_{i-1})}{1-\bar{\alpha}_i} x_i + \frac{\sqrt{\bar{\alpha}_{i-1}}\beta_i}{1-\bar{\alpha}_i}\hat{x}_0 + \tilde{\sigma}_i z$
    $x_{i-1} \leftarrow x_{i-1} - \zeta_i \nabla_{x_i} \log\left(\frac{1}{n}\sum_{j=1}^n \exp(-\ell_y(x_i^{(j)}))\right)$     $\triangleright x_i^{(j)}$ sampled *i.i.d.* from $\mathcal{N}(\hat{x}_0, \frac{\sigma_i^2}{1+\sigma_i^2}I)$
  **end for**
  **return** $\hat{x}_0$

---

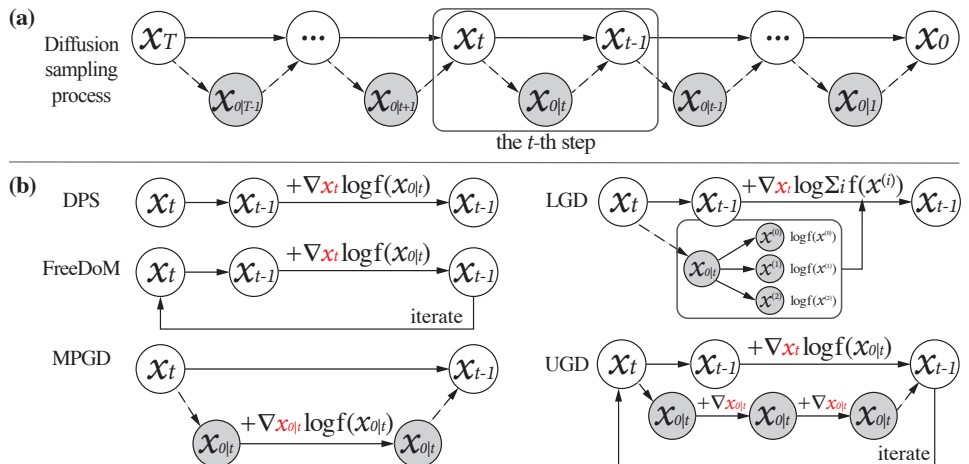

Figure 8: **(a)** The reversed diffusion process. **(b)** Illustration of different training-free guidance algorithms at the $t$-th reversed diffusion step.

# C Proofs

We prove Theorem 3.2 and Lemma 3.3 below.

## C.1 Proof of Theorem 3.2

*Proof.* For each algorithm, we prove the equivalence of design space separately below. Notice that when $\bar{\gamma} = 0$, $\tilde{f}$ degrades back to $f$.

**MPGD (Algorithm 3).** Below we demonstrate that any hyper-parameter $\{c_t\}_{t=1}^T$ in Algorithm 3 is equivalent to the TFG with $f(\boldsymbol{x}_{0|t}) = \exp\{-L(\boldsymbol{x}_{0|t}; y)\}$ and hyper-parameter

$$N_{recur} = 1, N_{iter} = 1, \bar{\gamma} = 0, \boldsymbol{\rho} = \boldsymbol{0}, \boldsymbol{\mu} = (c_1, \cdots, c_T)^\top.$$

To show this, notice that since both $\boldsymbol{\rho}$ and $\bar{\gamma}$ are zero, Line 4 and Line 7 take no effect. When using the identical sampling function (Line 9), TFG generates $\boldsymbol{x}_{t-1}$ using

$$\boldsymbol{x}_{t-1} = \text{Sample}(\boldsymbol{x}_t, \boldsymbol{x}_{0|t}, t) + c_t\sqrt{\bar{\alpha}_{t-1}}\nabla_{\boldsymbol{x}_{0|t}}\log f(\boldsymbol{x}_{0|t})$$

$$= \sqrt{\bar{\alpha}_{t-1}}\boldsymbol{x}_{0|t} + \sqrt{1 - \bar{\alpha}_{t-1} - \sigma_t^2}\frac{\boldsymbol{x}_t - \sqrt{\bar{\alpha}_t}\boldsymbol{x}_{0|t}}{\sqrt{1 - \bar{\alpha}_t}} + \sigma_t\epsilon_t + c_t\sqrt{\bar{\alpha}_{t-1}}\nabla_{\boldsymbol{x}_{0|t}}\log f(\boldsymbol{x}_{0|t})$$

$$= \sqrt{\bar{\alpha}_{t-1}}(\boldsymbol{x}_{0|t} - c_t\nabla_{\boldsymbol{x}_{0|t}}L(\boldsymbol{x}_{0|t}; y)) + \sqrt{1 - \bar{\alpha}_{t-1} - \sigma_t^2}\epsilon_\theta(\boldsymbol{x}_t, t) + \sigma_t\epsilon_t,$$

which is exactly the formula used in MPGD.

**DPS (Algorithm 2) and FreeDoM (Algorithm 4).** We prove both algorithms together as DPS is a special case of FreeDoM (without recurrence). Specifically, any hyper-parameter $\{\rho_t\}_{t=1}^T$, time travel step $r$ and distance function $\mathcal{D}(\boldsymbol{c}, \cdot)$ in Algorithm 4 is equivalent to TFG with $f(\boldsymbol{x}_{0|t}) = \exp\{-\mathcal{D}(\boldsymbol{c}, \boldsymbol{x}_{0|t})\}$ and hyper-parameter

$$N_{recur} = r, N_{iter} = 0, \bar{\gamma} = 0, \boldsymbol{\rho} = (\sqrt{\alpha_1}\rho_1, \cdots, \sqrt{\alpha_T}\rho_T)^\top, \boldsymbol{\mu} = \boldsymbol{0}.$$

To show this, notice that both algorithms have the identical resampling step from $\boldsymbol{x}_{t-1}$ to $\boldsymbol{x}_t$, so it suffices to prove that the formula to generate $\boldsymbol{x}_{t-1}$ in each recurrent step is the same. For FreeDoM, we have

$$\boldsymbol{x}_{t-1} = \text{Sample}(\boldsymbol{x}_t, \boldsymbol{x}_{0|t}, t) - \rho_t\nabla_{\boldsymbol{x}_t}\mathcal{D}(\boldsymbol{c}, \boldsymbol{x}_{0|t})$$

$$= \text{Sample}(\boldsymbol{x}_t, \boldsymbol{x}_{0|t}, t) - (\sqrt{\alpha_t}\rho_t)\nabla_{\boldsymbol{x}_t}\log f(\boldsymbol{x}_{0|t})/\sqrt{\alpha_t},$$

and the last line equals the combination of Line 7 and Line 7 in TFG.

**LGD (Algorithm 6).** Any hyper-parameter $\{\zeta_t\}_{t=1}^T, n$ in LGD is equivalent to TFG with $f(\boldsymbol{x}) = \exp\{-\ell_y(\boldsymbol{x})\}$, sample size $n$ in Line 4, and hyper-parameter

$$N_{recur} = 1, N_{iter} = 0, \bar{\gamma} = 1, \boldsymbol{\rho} = \boldsymbol{0}, \boldsymbol{\mu} = (\zeta_1, \cdots, \zeta_T)^\top.$$

Notice that $\bar{\alpha}_t$ in TFG equals $1/(1 + \sigma_t^2)$ in the DPS algorithm. With this, the equivalence is clear from the pseudo-code of both algorithms.

**UGD (Algorithm 5).** Any hyper-parameter $k, m, s(t)$ in UGD is equivalent to TFG with $f(\boldsymbol{x}) = \exp\{-\ell(c, \mathbf{f}(\boldsymbol{x}))\}$ and hyper-parameter

$$N_{recur} = k, N_{iter} = m, \bar{\gamma} = 0,$$

$$\boldsymbol{\rho} = (-\sqrt{\alpha_1}s(1)\delta_1, \cdots, -\sqrt{\alpha_T}s(T)\delta_T)^\top, \boldsymbol{\mu} = (-\sqrt{\frac{\alpha_1}{1 - \bar{\alpha}_1}}\delta_1, \cdots, -\sqrt{\frac{\alpha_T}{1 - \bar{\alpha}_T}}\delta_T)^\top,$$

where $\delta_t$ is the coefficient of $\epsilon_\theta(\boldsymbol{x}_t, t)$ in sampler $S$ in the UGD algorithm (which is negative). To show this, notice that $\Delta_t = \rho_t\nabla_{\boldsymbol{x}_t}\log\tilde{f}(\boldsymbol{x}_{0|t}) = -\rho_t\nabla_{\boldsymbol{x}_t}\ell(c, \mathbf{f}(\boldsymbol{x}_{0|t})) = \sqrt{\alpha_t}s(t)\delta_t\nabla_{\boldsymbol{x}_t}\ell(c, \mathbf{f}(\boldsymbol{x}_{0|t}))$. By replacing this into Line 9, the equivalence of guidance Variance Guidance can be easily observed. A similar deduction can be made for the mean guidance as well.

$\square$

## C.2 Proof of Lemma 3.3

*Proof.* Recall that $\boldsymbol{x}_{0|t} = \frac{\boldsymbol{x}_t - \sqrt{1-\bar{\alpha}_t}\epsilon_\theta(\boldsymbol{x}_t,t)}{\sqrt{\bar{\alpha}_t}}$. According to a simple chain rule, we have

$$\Delta_t = \rho_t \nabla_{\boldsymbol{x}_t} \log \tilde{f}(\boldsymbol{x}_{0|t}) \tag{9}$$

$$= \rho_t \frac{\boldsymbol{I} - \sqrt{1-\bar{\alpha}_t}\nabla_{\boldsymbol{x}_t}\epsilon_\theta(\boldsymbol{x}_t,t)}{\sqrt{\bar{\alpha}_t}}\nabla_{\boldsymbol{x}_{0|t}} \log \tilde{f}(\boldsymbol{x}_{0|t}). \tag{10}$$

The perfect optimization assumption implies the relationship between $\epsilon_\theta$ and the score of $p_t(\boldsymbol{x}_t)$, which we leverage and obtain

$$\Delta_t = \rho_t \frac{\boldsymbol{I} - \sqrt{1-\bar{\alpha}_t}\nabla_{\boldsymbol{x}_t}\epsilon_\theta(\boldsymbol{x}_t,t)}{\sqrt{\bar{\alpha}_t}}\nabla_{\boldsymbol{x}_{0|t}} \log \tilde{f}(\boldsymbol{x}_{0|t}) \tag{11}$$

$$= \rho_t \frac{\boldsymbol{I} + (1-\bar{\alpha}_t)\nabla^2_{\boldsymbol{x}_t}\log p_t(\boldsymbol{x}_t)}{\sqrt{\bar{\alpha}_t}}\nabla_{\boldsymbol{x}_{0|t}} \log \tilde{f}(\boldsymbol{x}_{0|t}). \tag{12}$$

Thus, it remains to draw a connection between the conditional covariance between $\Sigma_{0|t}$ and $\nabla^2_{\boldsymbol{x}_t}\log p_t(\boldsymbol{x}_t)$. Omit the subscript $\boldsymbol{x}_t$ in $\nabla_{\boldsymbol{x}_t}$, we have

$$\nabla^2 \log p_t(\boldsymbol{x}_t) = \frac{\nabla^2 p_t(\boldsymbol{x}_t)}{p_t(\boldsymbol{x}_t)} - \nabla \log p_t(\boldsymbol{x}_t)(\nabla \log p_t(\boldsymbol{x}_t))^\top \tag{13}$$

$$= \frac{1}{p_t(\boldsymbol{x}_t)} \int_{\boldsymbol{x}_0 \in \mathcal{X}} p_0(\boldsymbol{x}_0)\nabla^2 p_{t|0}(\boldsymbol{x}_t|\boldsymbol{x}_0)\mathrm{d}\boldsymbol{x}_0 - \nabla \log p_t(\boldsymbol{x}_t)(\nabla \log p_t(\boldsymbol{x}_t))^\top \tag{14}$$

$$\tag{15}$$

Notice that

$$\nabla^2 p_{t|0}(\boldsymbol{x}_t|\boldsymbol{x}_0) = p_{t|0}(\boldsymbol{x}_t|\boldsymbol{x}_0)\nabla^2 \log p_{t|0}(\boldsymbol{x}_t|\boldsymbol{x}_0) + \frac{1}{p_{t|0}(\boldsymbol{x}_t|\boldsymbol{x}_0)}\nabla p_{t|0}(\boldsymbol{x}_t|\boldsymbol{x}_0)(\nabla p_{t|0}(\boldsymbol{x}_t|\boldsymbol{x}_0))^\top$$

$$= -\frac{p_{t|0}(\boldsymbol{x}_t|\boldsymbol{x}_0)}{1-\bar{\alpha}_t}\boldsymbol{I} + p_{t|0}(\boldsymbol{x}_t|\boldsymbol{x}_0)(\frac{\boldsymbol{x}_t - \sqrt{\bar{\alpha}_t}\boldsymbol{x}_0}{1-\bar{\alpha}_t})(\frac{\boldsymbol{x}_t - \sqrt{\bar{\alpha}_t}\boldsymbol{x}_0}{1-\bar{\alpha}_t})^\top.$$

Replacing the LHS in the above equation in Equation (14), and noticing that $\mathbb{E}[\boldsymbol{x}_0|\boldsymbol{x}_t] = \frac{\boldsymbol{x}_t + (1-\bar{\alpha}_t)\nabla \log p_t(\boldsymbol{x}_t)}{\sqrt{\bar{\alpha}_t}}$, we have

$$\nabla^2 \log p_t(\boldsymbol{x}_t) = -\frac{1}{1-\bar{\alpha}_t}\boldsymbol{I} + \int_{\boldsymbol{x}_0 \in \mathcal{X}} p_{0|t}(\boldsymbol{x}_0|\boldsymbol{x}_t)(\frac{\boldsymbol{x}_t - \sqrt{\bar{\alpha}_t}\boldsymbol{x}_0}{1-\bar{\alpha}_t})(\frac{\boldsymbol{x}_t - \sqrt{\bar{\alpha}_t}\boldsymbol{x}_0}{1-\bar{\alpha}_t})^\top \mathrm{d}\boldsymbol{x}_0$$

$$- \nabla \log p_t(\boldsymbol{x}_t)(\nabla \log p_t(\boldsymbol{x}_t))^\top$$

$$= -\frac{1}{1-\bar{\alpha}_t}\boldsymbol{I} + \int_{\boldsymbol{x}_0 \in \mathcal{X}} p_{0|t}(\boldsymbol{x}_0|\boldsymbol{x}_t)(\frac{\boldsymbol{x}_t - \sqrt{\bar{\alpha}_t}\boldsymbol{x}_0}{1-\bar{\alpha}_t})(\frac{\boldsymbol{x}_t - \sqrt{\bar{\alpha}_t}\boldsymbol{x}_0}{1-\bar{\alpha}_t})^\top \mathrm{d}\boldsymbol{x}_0$$

$$- (\frac{\sqrt{\bar{\alpha}_t}\mathbb{E}[\boldsymbol{x}_0|\boldsymbol{x}_t] - \boldsymbol{x}_t}{1-\bar{\alpha}_t})(\frac{\sqrt{\bar{\alpha}_t}\mathbb{E}[\boldsymbol{x}_0|\boldsymbol{x}_t] - \boldsymbol{x}_t}{1-\bar{\alpha}_t})^\top$$

$$= -\frac{1}{1-\bar{\alpha}_t}\boldsymbol{I} + \mathrm{Cov}\Big[\frac{\sqrt{\bar{\alpha}_t}\boldsymbol{x}_0 - \boldsymbol{x}_t}{1-\bar{\alpha}_t}|\boldsymbol{x}_t\Big]$$

$$= -\frac{1}{1-\bar{\alpha}_t}\boldsymbol{I} + \frac{\bar{\alpha}_t}{(1-\alpha_t)^2}\Sigma_{0|t}.$$

The proof is finished by substituting $\nabla^2 \log p_t(\boldsymbol{x}_t)$ in Equation (12) by the above result. □

# D Task details

## D.1 Gaussian Deblur

In computer vision, the Gaussian deblur task addresses the challenge of restoring clarity to images that have been blurred by a Gaussian process. Gaussian blur, a common image degradation, simulates effects such as out-of-focus photography or atmospheric disturbances. It is characterized by the convolution of an image with a Gaussian kernel, a process that spreads the pixel values outwards, leading to a smooth, uniform blur [33]. The deblurring task seeks to reverse this effect, aiming to retrieve the original sharp image.

**Guidance target.** Specifically, we apply a $61 \times 61$ sized Gaussian blur with kernal intensity 3.0 to original $256 \times 256$ images. A random noise with a variance of $\sigma^2 = 0.05^2$ is added to the noisy images. If we denote the above process as a blurring operator $\mathcal{A}_{\text{blur}} : \boldsymbol{x} \to \boldsymbol{y}$, where $\boldsymbol{x} \in \mathbb{R}^{256 \times 256 \times 3}, \boldsymbol{y} \in \mathbb{R}^{256 \times 256 \times 3}$ are original images and noisy images, then the target of Gaussian Deblur is to generate a clean image $\boldsymbol{x}_0$ such that:

$$\max_{\boldsymbol{x}_0} p(\boldsymbol{x}_0) = \max_{\boldsymbol{x}_0} \exp(-\|\mathcal{A}_{\text{blur}}(\boldsymbol{x}_0) - \boldsymbol{y}\|_2),$$

which means if we project the generated image into the noisy space, the noisy samples $\mathcal{A}_{\text{blur}}(\boldsymbol{x})$ should be similar to the ground truth noisy images $\boldsymbol{y}$.

**Evaluation metrics.** In our experiments, we evaluate each guidance method on a set of 256 samples generated by Cat-DDPM. We apply the FID [26] to assess the fidelity, Learned Perceptual Image Patch Similarity (LPIPS) [80] to evaluate the guidance validity.

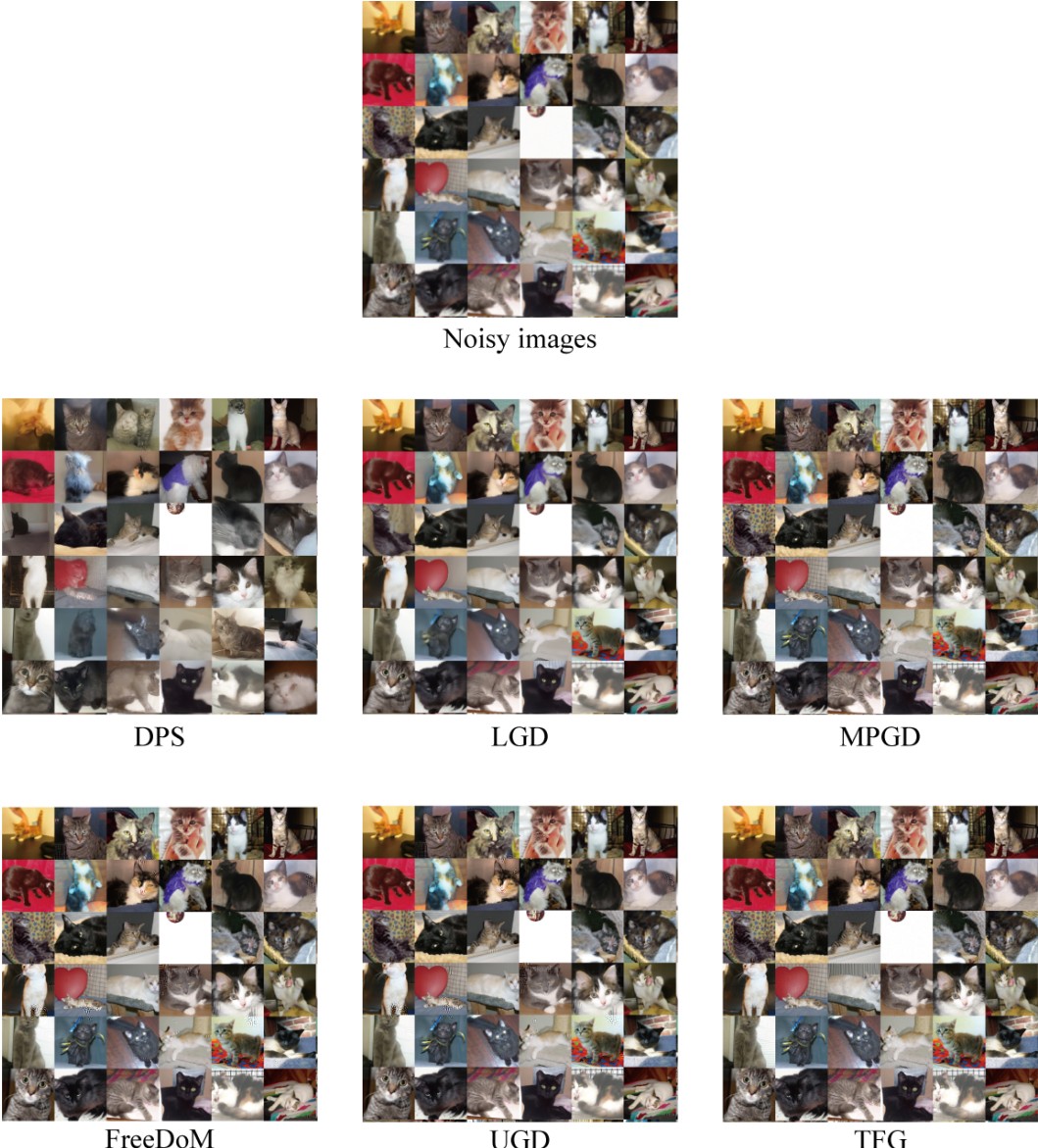

Figure 9: Quantitative comparison of different training-free guidance methods on Gaussian deblur task. Our TFG method can produce clean images without background noise (unlike FreeDoM and UGD), faithful image features (unlike DPS) and vivid image details (compared to LGD). $N_{\text{recur}}$ is set to 1 for all methods.

## D.2 Super Resolution

Super-resolution in computer vision refers to the process of enhancing the resolution of an imaging system, aimed at reconstructing a high-resolution image from one or more low-resolution observations. This technique is fundamental in overcoming the inherent limitations of imaging sensors and improving the detail and quality of digital images. Super-resolution has broad applications, ranging from satellite imaging and surveillance to medical imaging and consumer photography [46].

**Guidance target.** Specifically, we simply down-sample to original $256 \times 256$ images to the resolution of $64 \times 64$. A random noise with a variance of $\sigma^2 = 0.05^2$ is also added to the noisy images. If we denote the above process as a degradation operator $\mathcal{A}_{\text{down}} : \boldsymbol{x} \rightarrow \boldsymbol{y}$, where $\boldsymbol{x} \in \mathbb{R}^{256 \times 256 \times 3}, \boldsymbol{y} \in \mathbb{R}^{256 \times 256 \times 3}$ are original images and down-sampled images, then the target of super- is to generate a high resolution image $\boldsymbol{x}_0$ such that:

$$\max_{\boldsymbol{x}_0} p(\boldsymbol{x}_0) = \max_{\boldsymbol{x}_0} \exp(-\|\mathcal{A}_{\text{down}}(\boldsymbol{x}_0) - \boldsymbol{y}\|_2),$$

which means if we project the generated image into the downsampled image space, the downsampled samples $\mathcal{A}_{\text{down}}(\boldsymbol{x})$ should be similar to the ground truth downsampled images $\boldsymbol{y}$.

**Evaluation metrics.** Similar to Gaussian Deblur, we evaluate each guidance method on a set of 256 samples generated by Cat-DDPM. We apply FID [26] to assess the fidelity, Learned Perceptual Image Patch Similarity (LPIPS) [80] to evaluate the guidance validity.

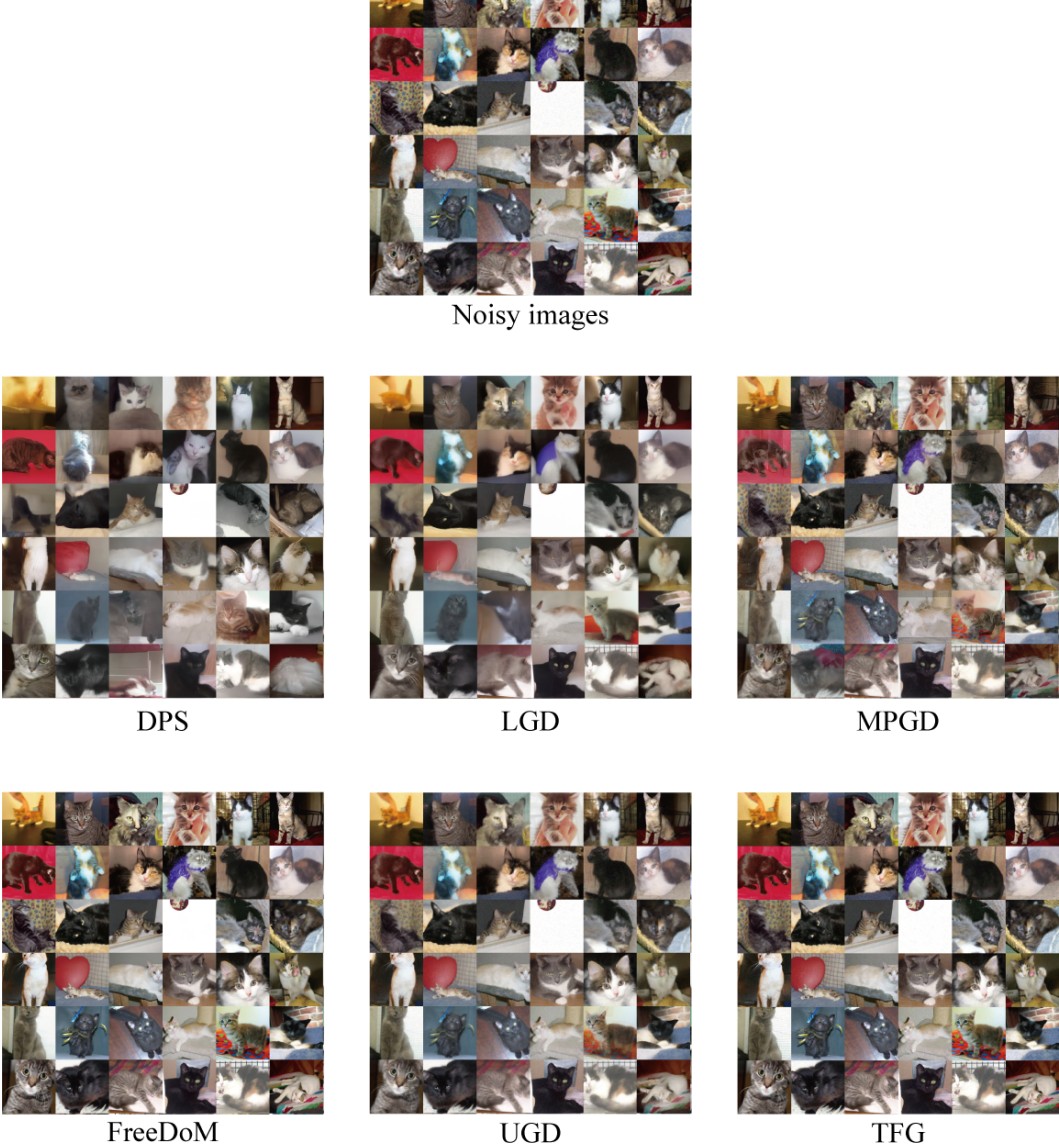

Figure 10: Quantitative comparison of different training-free guidance methods on super-resolution task. Our TFG method can produce clean images, faithful image features (unlike DPS, MPGD) and vivid image details (compared to LGD, UGD). $N_{\text{recur}}$ is set to 1 for all methods.

## D.3 Label Guidance

Label guidance is a standard task for conditional generation studied in previous literature [7, 23]. The target is to generate images conditioned on a certain label. We found this standard task is rarely studied in training-free guidance work and there exist an evident performance gap between training-based guidance and existing training-free guidance as shown in Section 3.

**Label sets.** In our experiments, we studied labels from CIFAR10 [30] and ImageNet [55]. We average the results on 10 labels from CIFAR10 if there is no extra explanation. For ImageNet, which is resource-hungry to do comprehensive inference, we randomly select 4 labels (111, 222, 333, 444) to evaluate the methods. The corresponding label-ID and their names are as follows:

Table 6: CIFAR-10 Dataset Labels

| Label-ID | Label Name |
|----------|------------|
| 0 | Airplane |
| 1 | Automobile |
| 2 | Bird |
| 3 | Cat |
| 4 | Deer |
| 5 | Dog |
| 6 | Frog |
| 7 | Horse |
| 8 | Ship |
| 9 | Truck |

Table 7: Selected ImageNet-1K Dataset Labels

| Label-ID | Label Name |
|----------|------------|
| 111 | nematode, nematode worm, roundworm |
| 222 | Kuvasz |
| 333 | Hamster |
| 444 | bicycle-built-for-two, tandem bicycle, tandem |

**Guidance target.** For each dataset, we use the output probability of a pre-trained classifier $h(\cdot)$ as the target probability. Our target is to maximize the certain classification probability of a given label, $i.e.$,

$$\max_{\boldsymbol{x}_0} p(\boldsymbol{x}_0) = \max_{\boldsymbol{x}_0} \text{softmax}(h(\boldsymbol{x}_0))_i,$$

where $i$ is the label-ID of the target label, and $h(\cdot)$ is the logits of $\boldsymbol{x}_0$ produced by the pre-trained classifier. For CIFAR10 and ImageNet, we use a pre-trained classifier based on ResNet [17] and VIT [9] that are provided from [75] and [9] respectively. The image resolution for CIFAR10 and ImageNet are $32 \times 32$ and $256 \times 256$ respectively.

**Evaluation metrics.** We follow the image generation literature to use Fréchet inception distance (FID) [20] to assess the fidelity of generated images. The reference images are filtered from the entire dataset of CIFAR10 or ImageNet with the target label, and we set sample sizes as 2560 and 256 for CIFAR and ImageNet respectively. For validity, we use another pre-trained classifier to compute accuracy other than the one used in providing guidance to avoid over-confidence:

$$accuracy = \frac{\#classified\ as\ target\ label}{\#generated\ samples}$$

For CIFAR10, we use a pre-trained ConvNeXT [36] downloaded from HuggingFace Hub[7]. And for ImageNet, we use the pre-trained DeiT [69] downloaded from HuggingFace Hub[8].

---

[7] https://huggingface.co/ahsanjavid/convnext-tiny-finetuned-cifar10
[8] https://huggingface.co/facebook/deit-small-patch16-224

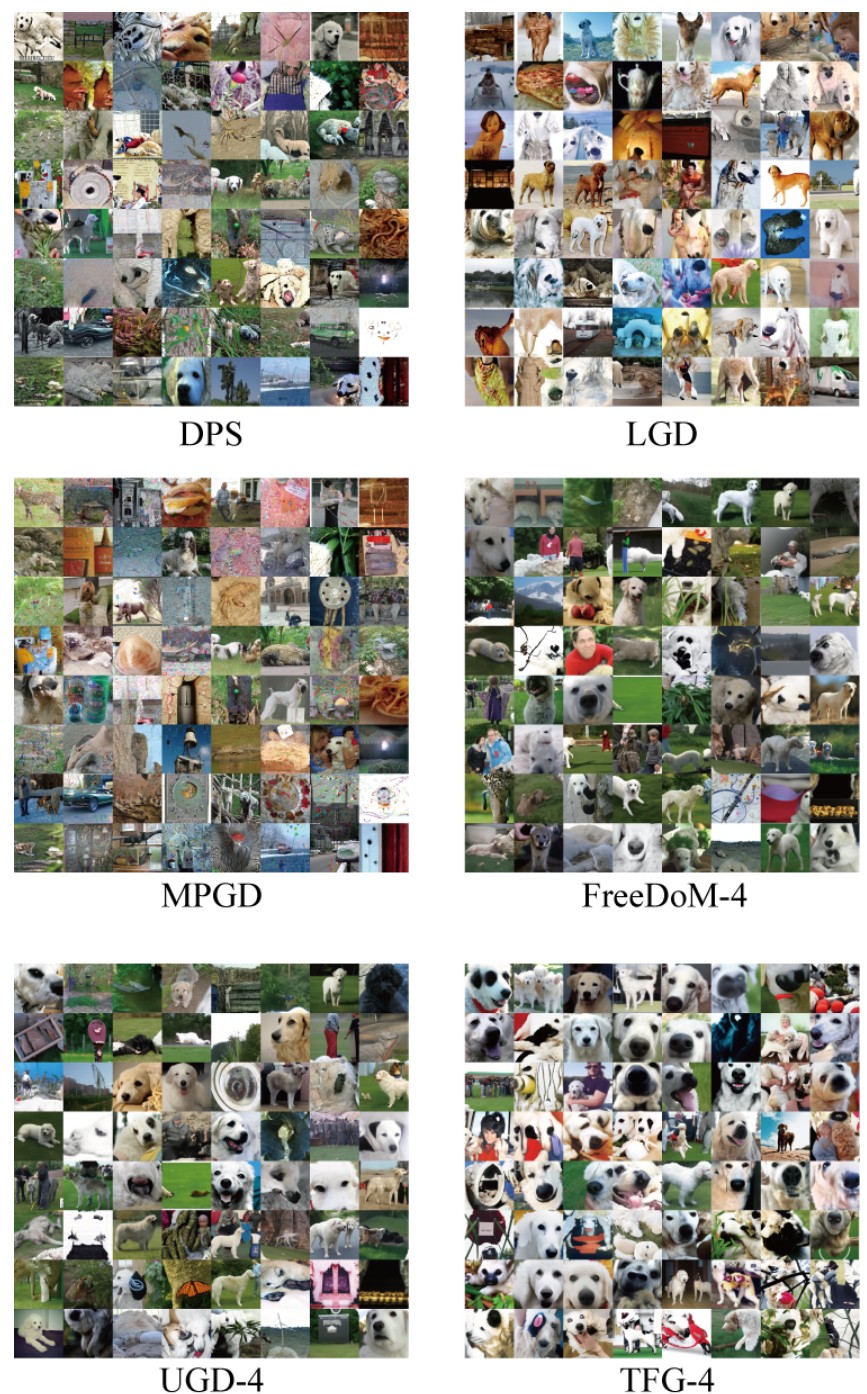

Figure 11: Quantitative comparison of different training-free guidance methods on ImageNet label guidance (with target = 222, Kuvasz). The suffix of FreeDoM, UGD, TFG represents the number of recurrence $N_{\text{recur}}$. Notice that all the samples are generated based on the same seed and we do not conduct cherry-picking. It is apprent that TFG generates the most valid samples among all the compared methods.

## D.4  Combined Guidance

An interesting scenario for conditional generation is to assign multiple targets for a single sample. Conditional generation with multiple conditions is crucial in machine learning for enhancing the relevance and applicability of AI across complex, real-world scenarios. It enables models to produce more contextually appropriate and personalized outputs, crucial for fields requiring high customization.

**Motivations.**   Combined guidance is to use multiple target functions to guide the same sample towards multiple targets for the same sample. It is more efficient for training-free guidance to do combined guidance as the space of combinatorial targets is potentially huge, which makes it unrealistic to train all target combinations for training-based guidance. We also find it related to the topic of spurious correlations [60], where certain combinations of attributes may dominant the other combinations. For example, hair color may have a strong correlation with gender in CelebA dataset [37]. It is beneficial to explore training-free guidance on reducing the bias of generation models trained on these biased data and address the concerns related to fairness and equality.

**Guidance target.**   We study combined guidance on CelebA-DDPM, which is trained on CelebA-HQ [26] dataset. The image resolution is $256 \times 256$. We choose two settings of combined guidance with two attributes: (gender, hair color) and (gender, age). Each attribute has two labels, where gender$\in \{$male, female$\}$, age$\in \{$young, old$\}$, and hair color$\in \{$black, blonde$\}$. We have a binary classifier for each attribute that is downloaded from HuggingFace Hub[9][10][11]. The target is to sample images that maximize the marginal probability:

$$\max_{\boldsymbol{x}_0} p_{\text{combined}}(\boldsymbol{x}_0) = \max_{\boldsymbol{x}_0} p_{\text{target1}}(\boldsymbol{x}_0) p_{\text{target2}}(\boldsymbol{x}_0),$$

where $p_{\text{target}}(\boldsymbol{x}_0)$ is computed using label guidance as shown in Appendix D.3.

**Evaluation metrics.**   As it is hard to filter many reference images for combined targets in CelebA-HQ dataset, we adopt Kernel Inception distance (KID) [5] to assess fidelity of generated samples using 1,000 random sampled images of CelebA-HQ as reference images. We generate 256 samples for each evaluated method. We follow MPGD [18] to use the logarithm of KID, $i.e.$ KID (log). For validity, we use another three attribute classifiers to compute the accuracy considering the conjunction of attributes:

$$accuracy = \frac{\# \wedge_{target\ label} \left(classified\ as\ target\ label\right)}{\# generated\ samples}$$

The evaluation classifiers are also downloaded from HuggingFace Hub[12][13][14].

---

[9]Age: https://huggingface.co/nateraw/vit-age-classifier
[10]Gender: https://huggingface.co/rizvandwiki/gender-classification-2
[11]Hair color: https://huggingface.co/enzostvs/hair-color
[12]Age (Evaluation): ibombonato/swin-age-classifier
[13]Gender (Evaluation): https://huggingface.co/rizvandwiki/gender-classification
[14]Age (Evaluation): https://huggingface.co/londe33/hair_v02

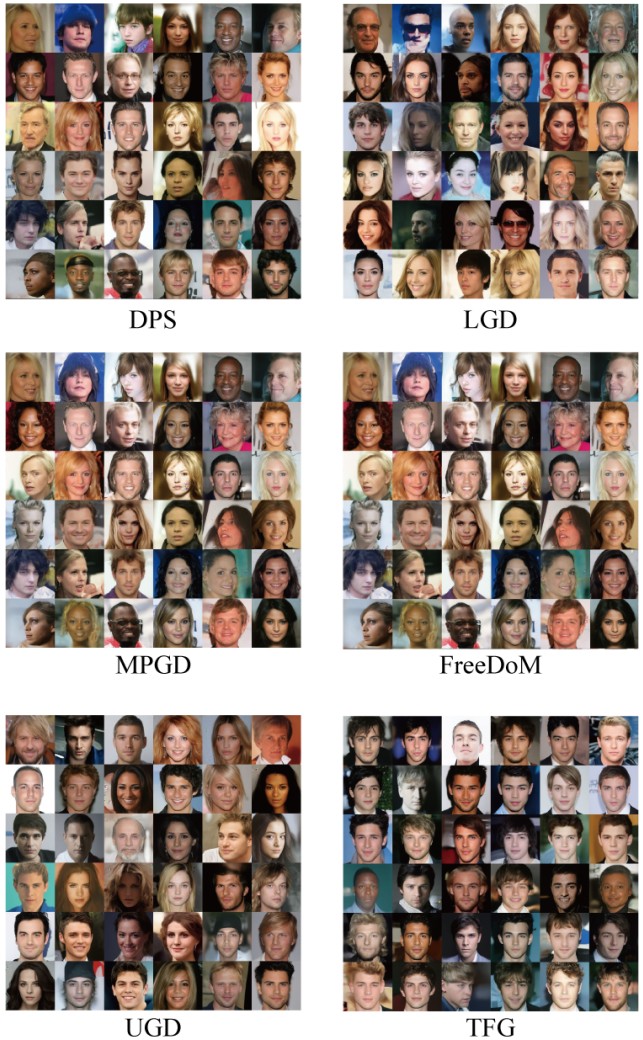

Figure 12: Quantitative comparison of different training-free guidance methods on combined guidance task (male+young). Our TFG method can produce images with high fidelity and validity compared to all the baselines. Notice that we use the fixed seed for all the methods in this figure and do not conduct cherry picking. $N_{\text{recur}}$ is set to 1 for all methods.

## D.5 Fine-grained Guidance

Fine-grained classification is a specialized task in computer vision that focuses on distinguishing between highly similar subcategories within a larger, general category. This task is particularly challenging due to the subtle differences among the objects or entities being classified. For example, in the context of animal species recognition, fine-grained classification would involve not just distinguishing a bird from a cat, but identifying the specific species of birds, such as differentiating between a crow and a raven [79].

Studying fine-grained generation in the context of generative models like Stable Diffusion or DALL-E presents unique challenges due to the inherent complexity of generating highly detailed and specific images. Fine-grained generation involves creating images that not only belong to a broad category but also capture the subtle nuances and specific characteristics of a narrowly defined subcategory. For example, generating images of specific dog breeds in distinct poses or environments requires the model to understand and replicate minute details that distinguish one breed from another.

**Motivations.**    To develop personalized AI, it is important to explore if the foundational generative models can synthesize fine-grained, accurate target samples according to user-defined target. However, this usually requires high-quality and detailed training data, and the model should be highly sensitive to small variations in input to accurately produce the desired output, which can be difficult for strong text2image generation models DALL-E[15] or Imagen[16]. We first study this problem in a training-free guidance scenario.

**Guidance target.**    We study the *out-of-distribution* fine-grained label guidance on ImageNet-DDPM, which learns the generation of some species of birds but not comprehensively. We use an EfficientNet trained to classify 525 fine-grained bird species downloaded from HuggingFace Hub.[17] The classifier is trained on Bird Species dataset on Kaggle[18]. We follow the same way in label-guidance to maximize softmax probability for target fine-grianed label. We randomly sample 4 labels (111, 222, 333, 444) in Bird Species dataset, which are:

Table 8: Selected Bird Species Dataset Labels

| Label-ID | Label Name |
| --- | --- |
| 111 | Brown headed cowbird |
| 222 | Fairy tern |
| 333 | Lucifer hummingbird |
| 444 | Scarlet macaw |

**Evaluation metrics.**    Similar to label guidance, we use FID to evaluate generation fidelity by filtering data of the target label as reference images. We also compute the FID with 256 sampled images. For accuracy, we adopt another downloaded pre-trained classifier trained on Bird Species dataset from HuggingFace Hub.[19]

---

[15]https://openai.com/index/dall-e-2/

[16]https://deepmind.google/technologies/imagen-2/

[17]https://huggingface.co/chriamue/bird-species-classifier

[18]https://www.kaggle.com/datasets/gpiosenka/100-bird-species/data

[19]https://huggingface.co/dennisjooo/Birds-Classifier-EfficientNetB2

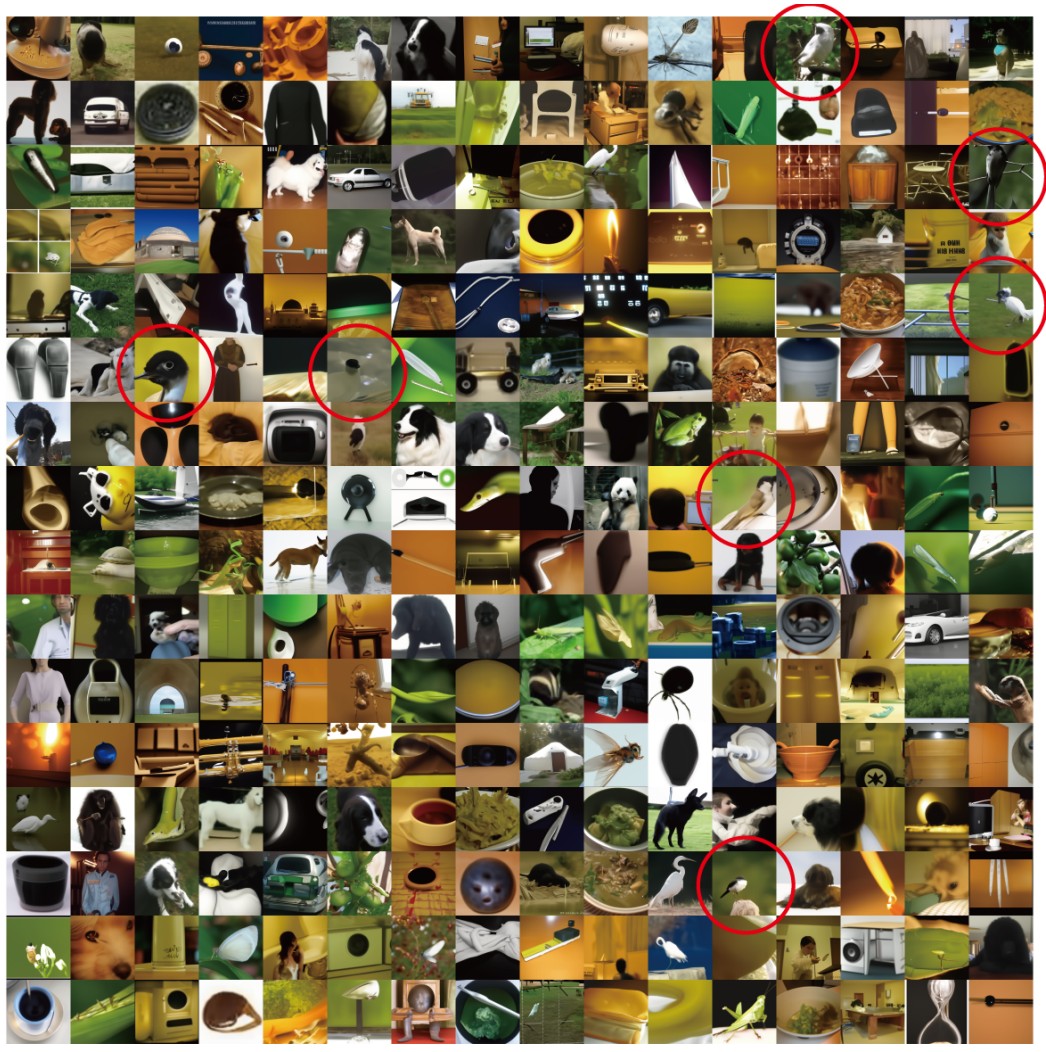

Figure 13: The sampled 256 images for fine-grained guidance with target label 222 (fairy tern) by ImageNet-DDPM with TFG. A key feature of fairy tern is its black-colored head. We observe that TFG generally samples images with black, round shapes and successfully generates some birds with target feature (red circled).

## D.6 Style Transfer

Style transfer is a significant task in computer vision (CV) that focuses on applying the stylistic elements of one image onto another while preserving the content of the target image. This task is pivotal because it bridges the gap between artistic expression and technological innovation, allowing for the creation of novel and aesthetically pleasing visual content. Applications of style transfer are vast, ranging from enhancing user engagement in digital media and advertising to aiding in architectural design by visualizing changes in real-time.

**Guidance target.** The target in our experiments is to guide a text-to-image latent diffusion model Stable-Diffusion-v-1-5 [53][20] to generate images that fit both the text input prompts and the style of the reference images. The guidance objective involves calculating the Gram matrices [25] of the intermediate layers of the CLIP image encoder for both the generated images and the reference style images. The Frobenius norm of these matrices serves as the metric for the objective function. Specifically, for a reference style image $x_{ref}$ and a decoded image $D(z_{0|t})$ generated from the

[20]https://huggingface.co/runwayml/stable-diffusion-v1-5

estimated clean latent variable $z_{0|t}$, we compute the Gram matrices $G(x_{\text{ref}})$ and $G(D(z_{0|t}))$. These matrices are derived from the features of the 3rd layer of the image encoder, in accordance with the methodologies described in MPGD [18] and FreeDoM [78]. The target function is computed as follows:

$$\max_{x_0} p_{\text{style}}(x_0) = \max_{x_0} \exp(-\|G(x_{\text{ref}}) - G(D(z_{0|t}))\|_F^2),$$

where $\|\cdot\|_F^2$ denotes the Frobenius norm of a matrix.

**Evaluation metrics.** We use Style score and CLIP score to assess the guidance validity and fidelity, respectively. For reference style images and text prompts, we select 4 images from WikiArt [56] that are also used by MPGD [18], and 64 text prompts from Partiprompts [77]. For each style, we generate 64 images based on the 64 different text prompts. To avoid over-confidence of CLIP score, we use two different CLIP models downloaded from HuggingFace Hub to compute guidance and evaluation metrics, respectively.[21][22] The style images and examplar prompts are shown as follows.

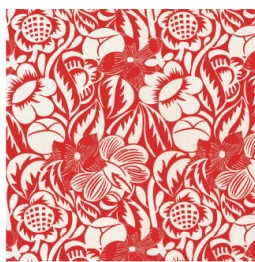 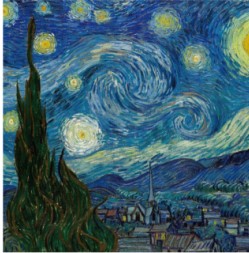 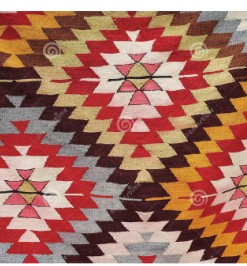 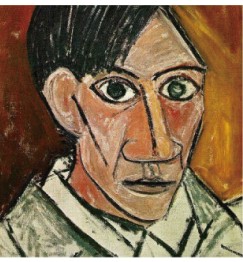

Figure 14: Four style images used in style transfer task.

Table 9: Examples for used prompts for style transfer tasks.

| Content Description | Content Description |
| --- | --- |
| A book with the words 'Don't Panic!' written on it | Ground view of the Great Pyramids and Sphinx on the moon's surface, Earth in the sky |
| A canal in Venice | A white towel |
| Portrait of a tiger wearing a train conductor's hat and holding a skateboard | Downtown Shanghai at sunrise, detailed ink wash |
| Background pattern with alternating roses and skulls | A smiling sloth holding a quarterstaff and a book, VW van parked on grass |
| A pickup truck | Concept of energy |
| h A shoe with a sock draped over it | A kitchen without a refrigerator |
| Times Square during the day | A squirrel |
| A turkey walking in the kitchen | A bowl |
| The Statue of Liberty in Minecraft | A man with wild hair looking into a crystal ball |
| Concentric circles | A fire hydrant with graffiti on it |

---

[21] Guidance: https://huggingface.co/openai/clip-vit-base-patch16
[22] Evaluation: https://huggingface.co/openai/clip-vit-base-patch32

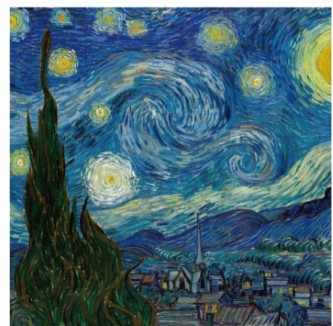

Target style

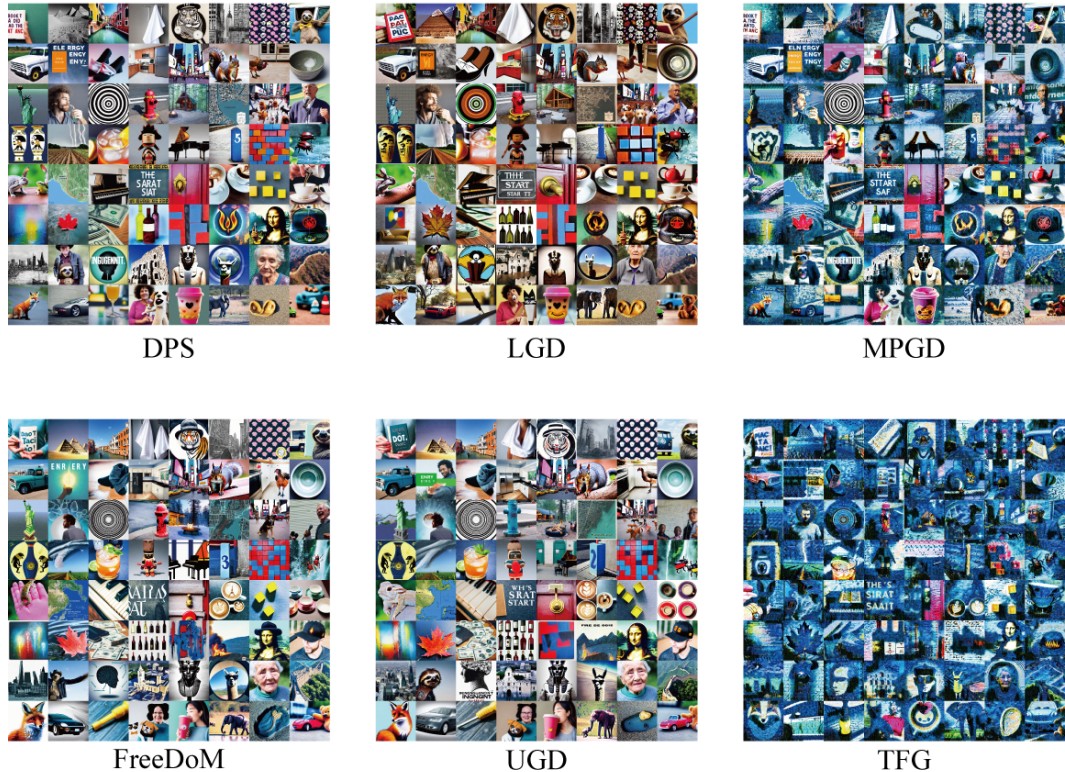

DPS         LGD         MPGD

FreeDoM        UGD         TFG

Figure 15: Quantitative comparison of different training-free guidance methods on style transfer task with the target image as *The Starry Night* by Von Gogh. Our TFG generates the images with the most faithful style, while DPS, LGD, FreeDoM, and UGD fail to capture the target style. The images of MPGD is also of good quality, but the style score is also inferior than TFG by a large margin. We set $N_{\text{recur}} = 1$ for all methods.

### D.7 Molecule Property Guidance

**Setup.** Our benchmark setup generally follows [24, 3] but with certain specifications to guarantee the overall framework abides by the training-free regime. For dataset, we employ QM9 [50] and adopt the split in [24] with 100,000 training samples. Following [24] and [3], the training set is further split into two halves that guarantees there is no data leakage. The first half is leveraged to train a property prediction network with EGNN [57] as the backbone, which serves as the ground truth oracle to provide the label used for MAE computation. We reuse the checkpoints released by [3] for the labeling network for all 6 properties. The second half is used to train the diffusion model as well as the guidance network. We adopt the unconditional generation version of EDM [24] as the diffusion model. The guidance network in general takes the same architecture as defined by [3] that, again, features EGNN as the backbone but outputs a single scalar as the predicted quantum mechanics property. The only difference lies in at training time we mask the diffusion time step by zeros and always use the original clean molecule structure as input, ensuring training-free objective. All the pretrained models are trained separately for different properties. At sampling time, we employ DDIM [62] sampler with 100 sampling steps, as opposed to [24, 3] that take 1000 sampling steps.

**Guidance target.** We study training-free guided generation of molecules on 6 quantum mechanics properties, including polarizability ($\alpha$), dipole moment ($\mu$), heat capacity ($C_v$), highest orbital energy ($\epsilon_{\text{HOMO}}$), lowest orbital energy ($\epsilon_{\text{LUMO}}$) and their gap ($\Delta\epsilon$). Denote the oracle property prediction network as $\mathcal{E}$. Our guidance target in this case is given by

$$f(\boldsymbol{x}, c) := \exp(-\|\mathcal{E}(\boldsymbol{x}) - c\|_2^2), \tag{16}$$

where $\boldsymbol{x}$ is the input molecule and $c$ is the target property value.

**Evaluation metrics.** We use Mean Absolute Error (MAE) and validity as our evaluation metrics. In particular, MAE is computed between the target value and the predicted value gathered from the labeling network. Validity is computed by RDKit which measures whether the molecule is chemically feasible. We generate 4096 molecules for each property for evaluation.

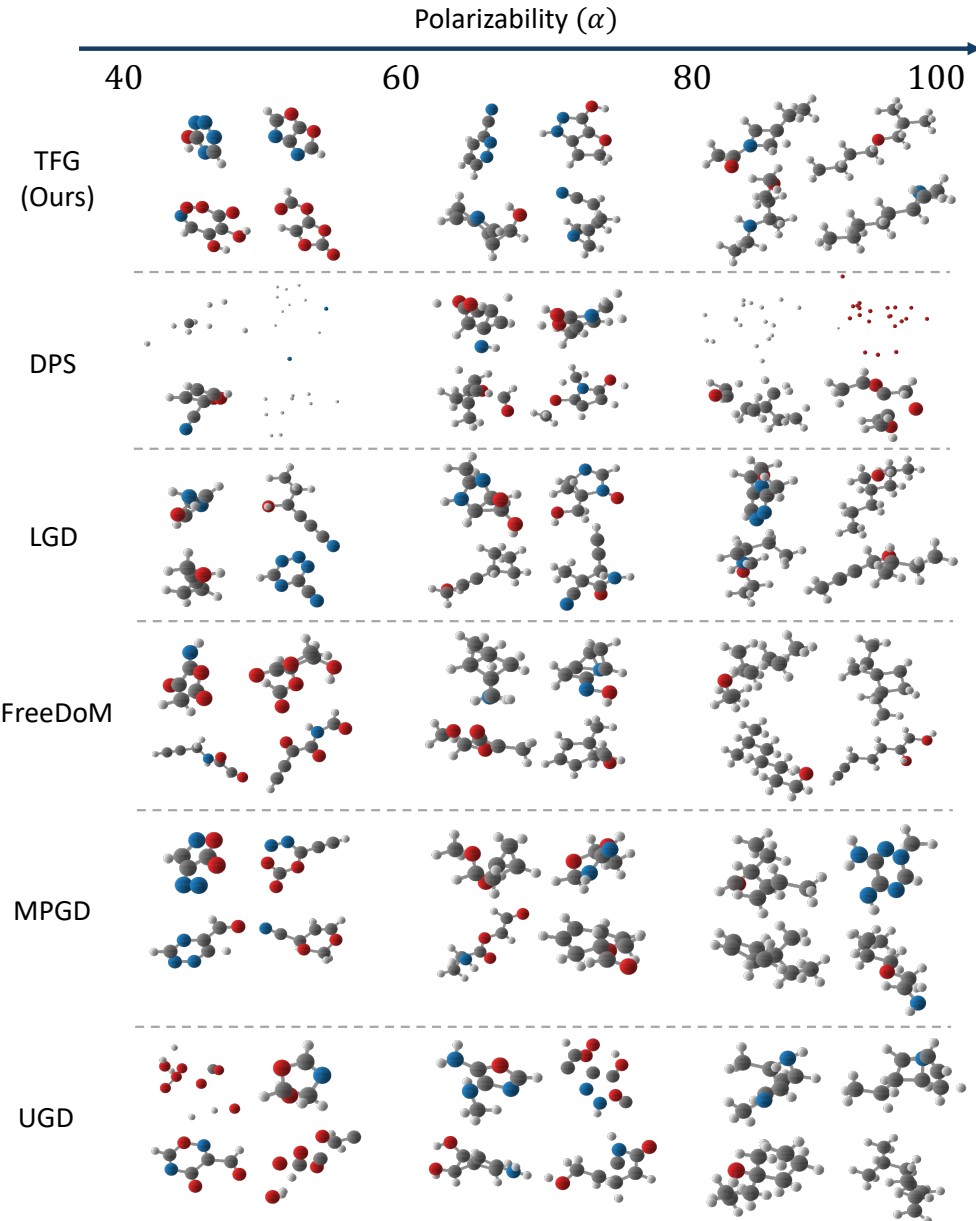

Figure 16: Quanlitative comparison of different training-free guidance methods on molecule generation task with the target property $\alpha$ (polarizability). Our TFG generates valid molecules with better design target, while baselines often fail to produce valid molecules or offer poor guidance towards the design target. The molecules generated by our approach are increasingly polarizable as $\alpha$ goes up.

## D.8 Audio Declipping

Audio declipping is a task in digital audio restoration where distorted audio signals are repaired. Clipping occurs when the amplitude or frequency of an audio signal exceeds the maximum limit that the recording system can handle, leading to a harsh, distorted sound with portions of the waveform "cut off." Declipping aims to reconstruct the missing parts of these clipped waveforms, restoring the audio's original dynamics and reducing distortion. This process improves the quality and clarity of the audio, making it more pleasant to listen to while preserving the original sound's integrity.

**Guidance target.**  Specifically, we apply a distortion operation to zero out the high frequency and low frequency (for the highest and lowest 40 dimensions) signal in the space of mel spectrograms. If we denote the above process as a blurring operator $\mathcal{A}_{\text{blur}} : \boldsymbol{x} \to \boldsymbol{y}$, where $\boldsymbol{x} \in \mathbb{R}^{256 \times 256}, \boldsymbol{y} \in \mathbb{R}^{256 \times 256}$ are mel spectrograms and noisy mel spectrograms for 5s audios, then the target of Audio Declipping is to generate a clean audio $\boldsymbol{x}_0$ such that:

$$\max_{\boldsymbol{x}_0} p(\boldsymbol{x}_0) = \max_{\boldsymbol{x}_0} \exp(-\|\mathcal{A}_{\text{blur}}(\boldsymbol{x}_0) - \boldsymbol{y}\|_2,$$

which means if we project the generate mel spectrogram into the noisy space, the noisy samples $\mathcal{A}_{\text{blur}}(\boldsymbol{x})$ should be similar to the ground truth noisy mel spectrogram $\boldsymbol{y}$.

**Evaluation metrics.**  In our experiments, we evaluate each guidance method on a set of 256 samples generated by Audio-diffusion. We apply the Dynamic time warping (DTW) [44] to assess the guidance validity, and Fréchet Audio Distance (FAD) [28] to assess the generation fidelity.

## D.9 Audio Inpainting

Audio inpainting is a digital audio restoration task that involves filling in missing or corrupted segments of an audio signal. Similar to image inpainting in the visual domain, this technique reconstructs the missing portions of sound by analyzing the surrounding context and seamlessly restoring the lost information. Applications of audio inpainting range from repairing damaged recordings to reconstructing gaps in audio streams due to data loss. The goal is to produce a natural-sounding result that preserves the continuity and overall quality of the original audio.

**Guidance target.** Specifically, we apply a distortion operation to zero out the middle 80 dimensions in the space of mel spectrograms. If we denote the above process as a blurring operator $\mathcal{A}_{\text{blur}} : \boldsymbol{x} \to \boldsymbol{y}$, where $\boldsymbol{x} \in \mathbb{R}^{256 \times 256}, \boldsymbol{y} \in \mathbb{R}^{256 \times 256}$ are mel spectrograms and noisy mel spectrograms for 5s audios, then the target of Audio Declipping is to generate a clean audio $\boldsymbol{x}_0$ such that:

$$\max_{\boldsymbol{x}_0} p(\boldsymbol{x}_0) = \max_{\boldsymbol{x}_0} \exp(-\|\mathcal{A}_{\text{blur}}(\boldsymbol{x}_0) - \boldsymbol{y}\|_2),$$

which means if we project the generate mel spectrogram into the noisy space, the noisy samples $\mathcal{A}_{\text{blur}}(\boldsymbol{x})$ should be similar to the ground truth noisy mel spectrogram $\boldsymbol{y}$.

**Evaluation metrics.** In our experiments, we evaluate each guidance method on a set of 256 samples generated by Audio-diffusion. We apply the Dynamic time warping (DTW) [44] to assess the guidance validity, and Fréchet Audio Distance (FAD) [28] to assess the generation fidelity.

# E  Experimental Details

## E.1  Details of Table 1

In Table 1, we study the effect of Monte-Carlo sample sizes in estimating the expectation of Line 4 in TFG algorithm. As the noise is added on both Mean Guidance ($\Delta_0$) and Variance Guidance ($\Delta_t$), we decouple the effect into adding noise solely on $\Delta_0$ (Mean only) or $\Delta_t$ (Variance only). In the setting of Variance only, we set $\boldsymbol{\mu} = 0, N_{\text{iter}} = 0, N_{\text{recur}} = 4, s_\rho(t) = $ "increase", and pick the best $\bar{\rho}$ and $\bar{\gamma}$ via hyper-parameter search. In the setting of Mean only, we set $\boldsymbol{\rho} = 0, N_{\text{iter}} = 4, N_{\text{recur}} = 1, s_\mu(t) = $ "increase", and pick the best $\bar{\mu}$ and $\bar{\gamma}$ via hyper-parameter search. We found that the sample size used in Monte-Carlo method play a neglect-able role on the performance if we set the optimal hyper-parameter. It is also noteworthy that the Monte-Carlo sampling does affect the performance of generated quality. For example, we can find that different targets shown in Appendix E.3 have different searched $\bar{\gamma}$. This indicates that the best $\bar{\gamma}$ for many targets are apparently not zero.

## E.2  Comparison with grid search

We compare the performance of our beam search parameters with the full grid search ones on CIFAR-10 label guidance task (Table 10). Overall, the performance of both search methods is identical, while grid search is much slower than our search strategy, indicating that our beam search strategy is effective and efficient.

Table 10: The searched $(\bar{\rho}, \bar{\mu}, \bar{\gamma})$ of exhaustive grid search and our beam search strategy on the CIFAR-10 label guidance task. We show the *validity* metric of the corresponding results and the gap $\Delta = \|validity_{\text{beam}} - validity_{\text{grid}}\|$. Overall, the performance of both methods is identical.

| Target | 0 | 1 | 2 | 3 | 4 | 5 | 6 | 7 | 8 | 9 | Avg. |
|---|---|---|---|---|---|---|---|---|---|---|---|
| $(\bar{\rho},\bar{\mu},\bar{\gamma})_{\text{grid}}$ | (1,2,0.001) | (0.25,2,0.001) | (2,0.25,1) | (4,0.5,0.1) | (1,0.5,0.001) | (2,0.25,0.001) | (0.25,0.5,1) | (1,0.5,0.001) | (1,0.25,0.001) | (0.5,2,0.001) | |
| $validity_{\text{grid}}$ | 80.44% | 35.38% | 28.25% | 56.32% | 29.57% | 41.70% | 52.66% | 42.14% | 83.35% | 73.22% | 52.30% |
| $(\bar{\rho},\bar{\mu},\bar{\gamma})_{\text{beam}}$ | (1,2,0.001) | (0.25,2,0.001) | (2,0.25,1) | (4,0.25,0.01) | (1,0.5,0.001) | (2,0.25,0.001) | (0.25,0.5,1) | (1,0.5,0.001) | (1,0.25,0.001) | (0.5,2,0.001) | |
| $validity_{\text{beam}}$ | 80.44% | 35.38% | 28.25% | 52.81% | 29.57% | 41.70% | 52.66% | 42.14% | 83.35% | 73.22% | 51.95% |
| $\Delta$ | 0.00% | 0.00% | 0.00% | 3.51% | 0.00% | 0.00% | 0.00% | 0.00% | 0.00% | 0.00% | 0.35% |

## E.3  Detailed results of each target and hyper-parameters

In this section, we present the hyper-parameters searched via the strategy introduced in Section 4 and the corresponding experimental results for TFG as shown in Table 11. We list several observations below.

- Overall, optimal parameters vary widely between problems and datasets. For example, even with the same model and objective (e.g., label classifier on ImageNet or CIFAR10), the best hyperparameters vary widely from target to target. This highlights the importance of hyperparameter search.

- The improvement of TFG over existing methods depends heavily on the difference between the optimal parameters and the subspaces of existing methods. For example, the $\bar{\rho}$ for UGD is the same as TFG for gender-age guidance task, where TFG only has 0.133% validity improvement over UGD. On the contrary, their values differ on the fine-grained classification task, and TFG has an 18.7% validity improvement over UGD. Overall, we suppose this depends on whether the optimal parameter lies in the subspace that existing methods can find.

- Though the baselines mentioned in our paper should be a special case of TFG, the results for the highest MO energy guidance in Table 3 show that MPGD outperforms TFG slightly. We want to point out that the reason TFG could occasionally have slightly worse performance in practice is due to the beam search computation limit we currently pose. More specifically, we allow TFG to search at most six steps (for all hyper-parameters) and all other methods for seven steps (in their subspaces). For the MO energy task, the searched parameter for MPGD is that $\bar{\mu} = 0.016$ (this is the only parameter that we need to search for MPGD), where the best (and last step) of TFG is that $\bar{\mu} = 0.004$ (because it uses one step to double another parameter). If we allocate more computational budget for the beam search steps, TFG will outperform MPGD on this target (in fact, eight steps suffice).

Table 11: The parameter $(\bar{\rho}, \bar{\mu}, \bar{\gamma})$ selected by beam search strategy for all methods, tasks, and targets. The search space of each method can be found in Section 3.1. For the detailed semantics of each task, please refer to Appendix D.

| | DPS | | | LGD | | | MPGD | | | FreeDoM | | | UGD | | | TFG | | |
|---|---|---|---|---|---|---|---|---|---|---|---|---|---|---|---|---|---|---|
| Target | $\bar{\rho}$ | $\bar{\mu}$ | $\bar{\gamma}$ | $\bar{\rho}$ | $\bar{\mu}$ | $\bar{\gamma}$ | $\bar{\rho}$ | $\bar{\mu}$ | $\bar{\gamma}$ | $\bar{\rho}$ | $\bar{\mu}$ | $\bar{\gamma}$ | $\bar{\rho}$ | $\bar{\mu}$ | $\bar{\gamma}$ | $\bar{\rho}$ | $\bar{\mu}$ | $\bar{\gamma}$ |
| *CIFAR-10 label guidance* | | | | | | | | | | | | | | | | | | |
| 0 | 1 | 0 | 0 | 16 | 0 | 1 | 0 | 2 | 0 | 1 | 0 | 0 | 2 | 2 | 0 | 1 | 2 | 0.001 |
| 1 | 8 | 0 | 0 | 16 | 0 | 1 | 0 | 4 | 0 | 0.5 | 0 | 0 | 4 | 4 | 0 | 0.25 | 2 | 0.001 |
| 2 | 1 | 0 | 0 | 16 | 0 | 1 | 0 | 0.25 | 0 | 1 | 0 | 0 | 0.25 | 0.25 | 0 | 2 | 0.25 | 1 |
| 3 | 4 | 0 | 0 | 8 | 0 | 1 | 0 | 8 | 0 | 2 | 0 | 0 | 1 | 1 | 0 | 4 | 0.25 | 0.01 |
| 4 | 0.5 | 0 | 0 | 2 | 0 | 1 | 0 | 0.25 | 0 | 0.5 | 0 | 0 | 4 | 4 | 0 | 1 | 0.5 | 0.001 |
| 5 | 4 | 0 | 0 | 0.25 | 0 | 1 | 0 | 0.25 | 0 | 1 | 0 | 0 | 4 | 4 | 0 | 2 | 0.25 | 0.001 |
| 6 | 1 | 0 | 0 | 4 | 0 | 1 | 0 | 0.5 | 0 | 16 | 0 | 0 | 4 | 4 | 0 | 0.25 | 0.5 | 1 |
| 7 | 2 | 0 | 0 | 0.5 | 0 | 1 | 0 | 0.5 | 0 | 0.5 | 0 | 0 | 4 | 4 | 0 | 1 | 0.5 | 0.001 |
| 8 | 2 | 0 | 0 | 16 | 0 | 1 | 0 | 2 | 0 | 1 | 0 | 0 | 4 | 4 | 0 | 1 | 0.25 | 0.001 |
| 9 | 4 | 0 | 0 | 0.5 | 0 | 1 | 0 | 2 | 0 | 1 | 0 | 0 | 4 | 4 | 0 | 0.5 | 2 | 0.001 |
| *ImageNet label guidance* | | | | | | | | | | | | | | | | | | |
| 111 | 2 | 0 | 0 | 2 | 0 | 1 | 0 | 8 | 0 | 1 | 0 | 0 | 8 | 8 | 0 | 2 | 0.5 | 0.1 |
| 222 | 2 | 0 | 0 | 2 | 0 | 1 | 0 | 0.25 | 0 | 0.5 | 0 | 0 | 2 | 2 | 0 | 0.5 | 1 | 0.1 |
| 333 | 2 | 0 | 0 | 2 | 0 | 1 | 0 | 0.25 | 0 | 0.25 | 0 | 0 | 8 | 8 | 0 | 1 | 4 | 1 |
| 444 | 4 | 0 | 0 | 4 | 0 | 1 | 0 | 4 | 0 | 1 | 0 | 0 | 4 | 4 | 0 | 0.5 | 2 | 0.1 |
| *Fine-grained guidance* | | | | | | | | | | | | | | | | | | |
| 111 | 0.25 | 0 | 0 | 0.25 | 0 | 1 | 0 | 0.25 | 0 | 0.25 | 0 | 0 | 0.25 | 0.25 | 0 | 0.5 | 0.5 | 0.01 |
| 222 | 0.25 | 0 | 0 | 1 | 0 | 1 | 0 | 0.25 | 0 | 0.5 | 0 | 0 | 4 | 4 | 0 | 0.5 | 0.5 | 0.01 |
| 333 | 0.25 | 0 | 0 | 0.25 | 0 | 1 | 0 | 0.5 | 0 | 0.25 | 0 | 0 | 4 | 4 | 0 | 0.5 | 0.5 | 0.01 |
| 444 | 0.25 | 0 | 0 | 0.25 | 0 | 1 | 0 | 0.25 | 0 | 1 | 0 | 0 | 1 | 1 | 0 | 0.5 | 0.5 | 0.01 |
| *Combined Guidance (gender & hair)* | | | | | | | | | | | | | | | | | | |
| (0,0) | 4 | 0 | 0 | 0.25 | 0 | 1 | 0 | 16 | 0 | 1 | 0 | 0 | 16 | 16 | 0 | 1 | 2 | 0.01 |
| (0,1) | 4 | 0 | 0 | 16 | 0 | 1 | 0 | 16 | 0 | 0.5 | 0 | 0 | 8 | 8 | 0 | 2 | 8 | 0.01 |
| (1,0) | 4 | 0 | 0 | 16 | 0 | 1 | 0 | 16 | 0 | 0.25 | 0 | 0 | 4 | 4 | 0 | 1 | 1 | 0.01 |
| (1,1) | 2 | 0 | 0 | 0.25 | 0 | 1 | 0 | 8 | 0 | 2 | 0 | 0 | 2 | 2 | 0 | 0.5 | 1 | 0.1 |
| *Combined Guidance (gender & age)* | | | | | | | | | | | | | | | | | | |
| (0,0) | 8 | 0 | 0 | 0.25 | 0 | 1 | 0 | 0.25 | 0 | 1 | 0 | 0 | 1 | 1 | 0 | 1 | 2 | 0.01 |
| (0,1) | 1 | 0 | 0 | 16 | 0 | 1 | 0 | 16 | 0 | 1 | 0 | 0 | 0.5 | 0.5 | 0 | 0.5 | 8 | 1 |
| (1,0) | 4 | 0 | 0 | 0.25 | 0 | 1 | 0 | 8 | 0 | 0.5 | 0 | 0 | 0.5 | 0.5 | 0 | 0.5 | 2 | 0.01 |
| (1,1) | 2 | 0 | 0 | 0.25 | 0 | 1 | 0 | 16 | 0 | 0.25 | 0 | 0 | 1 | 1 | 0 | 1 | 0.5 | 0.1 |
| *Super-resolution* | | | | | | | | | | | | | | | | | | |
| \ | 16 | 0 | 0 | 16 | 0 | 1 | 0 | 16 | 0 | 16 | 0 | 0 | 8 | 8 | 0 | 4 | 2 | 0.01 |
| *Gaussian Deblur* | | | | | | | | | | | | | | | | | | |
| \ | 16 | 0 | 0 | 16 | 0 | 1 | 0 | 16 | 0 | 16 | 0 | 0 | 16 | 16 | 0 | 1 | 8 | 0.01 |
| *Style Transfer* | | | | | | | | | | | | | | | | | | |
| 0 | 2 | 0 | 0 | 1 | 0 | 1 | 0 | 4 | 0 | 0.25 | 0 | 0 | 1 | 1 | 0 | 0.25 | 8 | 0.01 |
| 1 | 4 | 0 | 0 | 0.25 | 0 | 1 | 0 | 4 | 0 | 0.5 | 0 | 0 | 1 | 1 | 0 | 0.25 | 2 | 0.1 |
| 2 | 2 | 0 | 0 | 0.25 | 0 | 1 | 0 | 8 | 0 | 0.25 | 0 | 0 | 1 | 1 | 0 | 0.25 | 8 | 0.1 |
| 3 | 2 | 0 | 0 | 2 | 0 | 1 | 0 | 2 | 0 | 0.25 | 0 | 0 | 0.25 | 0.25 | 0 | 0.25 | 8 | 0.01 |
| *Molecule Property* | | | | | | | | | | | | | | | | | | |
| $\alpha$ | 0.005 | 0 | 0 | 0.005 | 0 | 1 | 0 | 0.01 | 0 | 0.01 | 0 | 0 | 0.02 | 0.02 | 0 | 0.016 | 0.001 | 0.0001 |
| $\mu$ | 0.02 | 0 | 0 | 0.01 | 0 | 1 | 0 | 0.005 | 0 | 0.02 | 0 | 0 | 0.005 | 0.005 | 0 | 0.001 | 0.002 | 0.1 |
| $C_v$ | 0.005 | 0 | 0 | 0.005 | 0 | 1 | 0 | 0.005 | 0 | 0.005 | 0 | 0 | 0.005 | 0.005 | 0 | 0.004 | 0.001 | 0.001 |
| $\epsilon_{\text{HOMO}}$ | 0.005 | 0 | 0 | 0.005 | 0 | 1 | 0 | 0.01 | 0 | 0.005 | 0 | 0 | 0.005 | 0.005 | 0 | 0.002 | 0.004 | 0.001 |
| $\epsilon_{\text{LUMO}}$ | 0.005 | 0 | 0 | 0.01 | 0 | 1 | 0 | 0.01 | 0 | 0.005 | 0 | 0 | 0.005 | 0.005 | 0 | 0.016 | 0.002 | 0.0001 |
| $\Delta$ | 0.005 | 0 | 0 | 0.01 | 0 | 1 | 0 | 0.01 | 0 | 0.01 | 0 | 0 | 0.005 | 0.005 | 0 | 0.032 | 0.001 | 0.001 |
| *Audio Declipping* | | | | | | | | | | | | | | | | | | |
| \ | 1 | 0 | 0 | 16 | 0 | 1 | 0 | 16 | 0 | 4 | 0 | 0 | 4 | 4 | 0 | 1 | 1 | 0.1 |
| *Audio Inpainting* | | | | | | | | | | | | | | | | | | |
| \ | 16 | 0 | 0 | 16 | 0 | 1 | 0 | 16 | 0 | 4 | 0 | 0 | 16 | 16 | 0 | 0.25 | 2 | 0.1 |

## E.4 Tricks implemented in FreeDoM codebase

In the codebase of FreeDoM[23], the schedule of guidance strength for different applications is different. For example, the guidance strength has a schedule coefficient $\sqrt{\bar{\alpha}_t}$ for face generation, and the schedule for style transfer is complex and involves a correction term, the mean of gradients' norm, and a specific constant coefficient 0.2. The paper does not mention this particular schedule, leaving

---

[23]https://github.com/vvictoryuki/FreeDoM

the rationale for choosing these schedules unclear. We choose not to include the tricks and find that with our unified hyper-parameter searching strategy, the performance of FreeDoM is similar.

## E.5 Hardware and Software

We run most of the experiments on clusters using NVIDIA A100s. We implemented our experiments using PyTorch [49] and the HuggingFace library.[24] Overall, we estimated that a total of 2,000 GPU hours were consumed.

---

[24]https://huggingface.co/

