# OpenReview forum: "TFG: Unified Training-Free Guidance for Diffusion Models"
_NeurIPS.cc/2024/Conference — NeurIPS 2024 spotlight_

### Official Review · Reviewer_Xm6V · 2024-06-17

**Soundness:** 3
**Presentation:** 4
**Contribution:** 4
**Rating:** 8
**Confidence:** 4

**Summary:**

The paper introduces Training-Free Guidance (TFG), a novel framework designed to enhance the generation of samples with desired properties using diffusion models, without necessitating additional model training. TFG aims to resolve the shortcomings of existing training-free methods by offering a unified algorithmic framework that simplifies the comparison and application of such methods across a wide range of tasks. By theoretically and empirically analyzing a hyper-parameter design space within this framework, the authors develop an effective strategy for hyper-parameter selection applicable to various tasks. Their comprehensive benchmarks across multiple tasks and targets demonstrate TFG's superior performance, achieving an average improvement of 7.4% over existing methods.

**Strengths:**

1. Given the burgeoning interest in the area of training-free guidance, the establishment of a unified benchmark as presented in this paper is a commendable contribution that holds the potential to significantly advance research in this field. The authors' efforts in this direction are highly appreciated and underscore the importance of standardized benchmarks for facilitating future developments.

2. The experiments conducted in this study are extensive, reflecting a high degree of diligence and thoroughness. Such a comprehensive experimental approach is commendable and warrants recognition. Consequently, I believe this aspect of the paper merits a positive evaluation for its contribution to validating the proposed framework and its applicability across a diverse range of tasks.

**Weaknesses:**

There are several areas where improvements could significantly enhance its contributions. Addressing these points satisfactorily would make a strong case for elevating the paper's status to "Accept" or "Strong Accept".

1. While the paper provides valuable insights into a specific line of training-free guidance, it is important to acknowledge the existence of a broader spectrum of works outside this category. This observation suggests that the title of the paper may slightly overreach, potentially implying a more comprehensive coverage than is actually presented. (Question 1)

2. The paper commendably supports the motivations behind conditional diffusion in Appendix A, offering a solid foundation for its relevance. However, the rationale for focusing specifically on training-free approaches appears less extensively articulated. (Question 2)

3. The authors claimed that they have theoretically grounded their unified framework. However, upon review, the theoretical underpinnings presented seem to require further elaboration to fully substantiate this claim. (Question 3, 4, 5)

4. The authors claimed that "the studies of training-free methods become the study within the hyperparameter space of our framework". Nonetheless, it appears that the search space defined within this framework might not fully encompass the range of hyper-parameters considered in existing literature. This limitation could potentially restrict the framework's applicability or comparative analysis capabilities. (Question 6, 7)

**Questions:**

1. The field of training-free guidance encompasses a wide array of studies beyond those specifically addressed in this paper (e.g. [1]-[3]). Could you elaborate on how these works relate to the scope and contributions of your paper?

2. Your paper operates under the assumption that the forward model is known, leveraging training-free guidance for solving inverse problems. Given that knowing the forward model allows for the generation of a large number of samples to train a conditional diffusion model at a relatively low computational cost (e.g., 10 A100 GPU hours) and training-based approaches are much better than training-free ones, could you discuss the significant motivations of adopting a training-free approach in this context?

3. The theoretical foundation of your paper seems to rest significantly on Lemma 4.1, which revisits the variance of MMSE estimator of the signal corrupted by Gaussian noise (e.g., (2.8) in [4]). This formula is widely adopted in diffusion papers (e.g., [5]). Given its established nature and previous applications, could you elaborate on how this lemma specifically contributes to the novel aspects of your framework?

4. Concerning Lemma 4.1, it appears there is no assurance that the generated image accurately follows the conditional distribution, nor is there a guarantee that the loss decreases at each iteration as suggested by equation (7). Could you provide further clarification or additional theoretical support to address these concerns?

5. Several techniques introduced in the paper lack direct theoretical underpinning:
- The concept of "time-travel" being an Ornstein-Uhlenbeck process is intriguing but lacks a detailed derivation. Could you expand on how the theory of the Ornstein-Uhlenbeck process quantifies the benefits of time-travel in your framework?
- The selection of hyperparameters seems not to be grounded in theory. Could you discuss the rationale behind these choices and any potential theoretical support?

6. The paper posits that the study of training-free methods can be encapsulated within the hyperparameter space of your framework. However, specific hyperparameter settings critical for the performance in existing works, such as the step sizes used in Face generation (at.sqrt()) and Style transfer ((correction * correction).mean().sqrt().item() * unconditional_guidance_scale / (norm_grad * norm_grad).mean().sqrt().item() * 0.2) in FreeDoM, are not explicitly covered. Could you address the omission of these settings and their impact on the comprehensiveness of your study?

7. The hyperparameter search settings for the baselines in your comparative analysis are not disclosed, raising questions about the fairness and validity of the comparisons. Could you provide more details on these settings to ensure a transparent and equitable comparison?

[1] Feng, Weixi, et al. "Training-free structured diffusion guidance for compositional text-to-image synthesis." ICLR 2023.

[2] Chen, Minghao, et al. "Training-free layout control with cross-attention guidance." WACV 2024.

[3] Mo, Sicheng, et al. "Freecontrol: Training-free spatial control of any text-to-image diffusion model with any condition." CVPR 2024.

[4] Efron, Bradley. "Tweedie’s formula and selection bias." JASA 2011.

[5] Kadkhodaie, Zahra, et al. "Generalization in diffusion models arises from geometry-adaptive harmonic representation." ICLR 2024.

**Limitations:**

Yes

---

> ### Author Rebuttal · Authors · 2024-08-07
>
> We sincerely thank you for your insightful and constructive review, and we are so honored that you believe our work is novel, effective, comprehensive, and offers commendable contributions. We are delighted to address your concerns and questions below.
>
> 1. The existence of a broader spectrum of works and the overreach of the title.
>
> > The field of training-free guidance encompasses …a wide array of studies beyond those specifically addressed in this paper (e.g. [1]-[3]). Could you elaborate on how these works relate to the scope and contributions of your paper?
> >
>
> Thanks for this valuable suggestion. The three works you mentioned are training-free methods specifically for text-to-image tasks (compositional generation [1], layout control [2], spatial control [3]). [1] and [2] exploited the cross-attention mechanism in Stable Diffusion UNet to incorporate guidance targets (noun phrases [1], layout condition [2]), while [3] leverages the diffusion features of the guidance image (with DDIM inversion) to guide the diffusion process.
>
> Unlike these works, which design task-specific guiding strategies, our paper focuses on general training-free guidance methods that can be applied to any diffusion model for any objective function. Basically, our work doesn’t focus on the design of the objective function for each task but on the universal methodology of guidance. That said, the improvement in architecture, diffusion models, and guidance targets is “orthogonal” to our work. We would like to discuss these works in our revised manuscript to clarify our scope and contributions.
>
> 2. The rationale for focusing specifically on TFG.
>
> > Your paper operates under the assumption that the forward model is known, leveraging training-free guidance for solving inverse problems. Given that knowing the forward model allows for the generation of a large number of samples to train a conditional diffusion model at a relatively low computational cost (e.g., 10 A100 GPU hours) and training-based approaches are much better than training-free ones, could you discuss the significant motivations of adopting a training-free approach in this context?
> >
>
> We sincerely thank the question that allows us to discuss the motivation of the paper further. In lots of cases when the conditions are complex, e.g. requires an image of a special dog species in certain scenarios or a molecule that has certain polarizability and synthesizability, they are extremely rare (with probability less than, e.g., $10^{-6}$) across the unconditional distribution. As such, “sampling a large number of targets for training a time-dependent classifier” itself is already impossible. However, training-free methods can effectively increase the sample rate from $10^{-6}$ to $10^{-2}$, making it possible to leverage training-based methods. In fact, we believe that one motivation of TFG is exactly to bridge towards the scenarios you describe. We will explicitly add this discussion in the paper.
>
> 3. Further elaboration of the theoretical part of the work.
>
> > The theoretical foundation of your paper seems to rest significantly on Lemma 4.1, which revisits the variance of MMSE estimator of the signal corrupted by Gaussian noise (e.g., (2.8) in [4]). This formula is widely adopted in diffusion papers (e.g., [5]). Given its established nature and previous applications, could you elaborate on how this lemma specifically contributes to the novel aspects of your framework?
> >
>
> We want to highlight that lemma 4.1 is simply used to illustrate the effect of “variance guidance” and is not used to prove our major theorem 3.2. Despite its wide use in previous works, here we use the lemma to point out that $\Delta _t$ in Line 7 is to control the second-order information, pointing to the difference between $\Delta_t$ and $\Delta_0$ from a theoretical perspective.
>
> > Concerning Lemma 4.1, it appears there is no assurance that the generated image accurately follows the conditional distribution, nor is there a guarantee that the loss decreases at each iteration as suggested by equation (7). Could you provide further clarification or additional theoretical support to address these concerns?
> >
>
> We agree that a theoretical guarantee on loss is important, but to the best of our knowledge, none of the existing training-free methods can prove that the generated sample follows the correct distribution as desired. This is not an issue of Lemma 4.1, but we believe that one crucial future direction is to provide a global-level guarantee for training-free guidance. The intrinsic difficulty here is to analyze the difference between a training-based classifier $f(x,t)$ and standard classifier $f(x)$, which are complex to be quantitatively captured.
>
> > The concept of "time-travel" being an Ornstein-Uhlenbeck process is intriguing but lacks a detailed derivation. Could you expand on how the theory of the Ornstein-Uhlenbeck process quantifies the benefits of time-travel in your framework?
> >
>
> We are delighted to further explain this. For each recurrent step, we compute $x_{t-1} = u(x_t)$ where $u$ corresponds to Line 6-9 and then add a Gaussian noise back in Line 10 to obtain updated $x_t$. Together, the process becomes
>
> $$
> d x_t = \sqrt{\alpha_t }u(x_t) - x_t + \sqrt{1 - \alpha_t} \epsilon,
> $$
>
> which is a typical OU process. This implies that when $N_{recur}$ goes to infinity, $x_t$ will converge to a certain distribution that is hard to compute analytically.

---

> ### Author Response · Authors · 2024-08-07
> **Additional responses**
>
> > The selection of hyperparameters seems not to be grounded in theory. Could you discuss the rationale behind these choices and any potential theoretical support?
> >
>
> Our beam-search hyperparameter selection strategy depends on an intuition (assumption) that when fixing all other parameters, the loss function of the remaining parameter is a decrease-then-increase function, i.e., as the parameter increases from 0 to infinity, the loss first goes down (or never goes down) and then increases. This is observed and summarized from experiments, and under such an assumption, it can be proved that the (near-)optimal hyperparameter can be found with sufficient search step and small enough stepping size (multiplying factor). We would add this to our paper as suggested.
>
> 4. The coverage of the proposed space in existing literature.
>
> > The paper posits that the study of training-free methods can be encapsulated within the hyperparameter space of your framework. However, specific hyperparameter settings critical for the performance in existing works, such as the step sizes used in Face generation (at.sqrt()) and Style transfer ((correction * correction).mean().sqrt().item() * unconditional_guidance_scale / (norm_grad * norm_grad).mean().sqrt().item() * 0.2) in FreeDoM, are not explicitly covered. Could you address the omission of these settings and their impact on the comprehensiveness of your study?
> >
>
> We are delighted to explain this further. Theoretically speaking, TFG aims to provide a general framework that unifies different sub-hyperparameter spaces, and the practical implementation tricks are omitted (as this trick only appears in FreeDoM code, not its original paper). That said, our framework can naturally represent this nuanced setting. For example, if we want the gradient to be $\frac{\nabla g}{\| \nabla g\|}$ for a given target function $g$, we could simply set the $f$ in our algorithm as $\exp\{\int \frac{\mathrm dg}{\| \nabla g\|}\}$, in which case the gradient of $f$ becomes exactly what we want.
>
> In practice, however, we observe that with the aid of hyperparameter searching, the output quality of our FreeDoM algorithm is comparable to the original implementation. Because of this, and to be consistent with the paper, we simply focus on the setting without normalization. Overall, we believe that such normalization should be similar to normalizing $\mathbf \rho$.
>
> > The hyperparameter search settings for the baselines in your comparative analysis are not disclosed, raising questions about the fairness and validity of the comparisons. Could you provide more details on these settings to ensure a transparent and equitable comparison?
> >
>
> We apologize for not providing a more comprehensive discussion beyond what was presented in Appendix E. We'd like to offer additional details here. For all methods discussed in our paper, we employed an identical beam search strategy with three beam search trials and a maximum of 7 steps. For each search, we start with initial parameters (identical across all methods) and double parameters to see if performance improved. Then, we keep the top 3 performances in the beam list after each step until the list becomes unchanged or the maximum step is reached. This beam search program is method-agnostic and compatible with all methods, ensuring fair and objective comparisons. To promote transparency and facilitate future research, we will open-source our code, all beam search runs, the best parameters for each algorithm, and their corresponding performance results.
>
> Again, we sincerely appreciate your helpful review. Please let us know whether there is any additional concern that we could address to improve your evaluation of our work.

---

> ### Comment · Reviewer_Xm6V · 2024-08-08
>
> I appreciate the authors' effort in addressing my concerns, which has led me to moderately increase the evaluation score.
>
> Regarding Question 2, I have experimented with several methods to fine-tune a network trained on clean images for use with noisy images [A, B]. My findings indicate that the training can be completed in one A100 hour and the performance is quite satisfactory. Could you please provide further clarification on this point?
>
> For Question 3, you assert that Theorem 3.2 represents the main theoretical contribution. However, it appears to be straightforward as the proposed method is a combination of these methods. Could you explain why the proof is non-trivial?
>
> In relation to Question 5, I would be interested in a more detailed discussion on the practical implications of the connections between the Ornstein-Uhlenbeck process and time travel. Specifically, how do these connections yield new guarantees or insights that could potentially enhance the methodology of time travel?
>
> [A] More Control for Free! Image Synthesis with Semantic Diffusion Guidance
>
> [B] Towards practical plug-and-play diffusion models

---

> > ### Author Response · Authors · 2024-08-09
> > **Response to the followup questions**
> >
> > We thank the reviewer for the moderately improved evaluation and we are delighted to address the follow-up concerns.
> >
> > > Regarding Question 2, I have experimented with several methods to fine-tune a network trained on clean images for use with noisy images [A, B]. My findings indicate that the training can be completed in one A100 hour and the performance is quite satisfactory. Could you please provide further clarification on this point?
> > >
> >
> > We have gone through both papers you provided, and we believe that there are major difference between our methods and theirs. Overall, the training efficiency of their method (even if we don’t think about the advantage of “training-free” at all) is highly determined by the structure of the guidance function, and the situation of using CLIP for guidance might be completely different from that of using a general classifier.
> >
> > For example, notice that in [A] the guidance function $F(x_t, t,l)$ is expected to have a special structure, i.e., $F(x_t, t, l) = E(x_t,t) \cdot E(l)$, where $l$ is the language embeddings and $x_t$ is the noisy images. In such case, using undesired images generated from unconditional diffusion models to help learn $E(x_t, t)$ from CLIP model $E(x_0)$ is likely **generalizable** to desired images since the relationship between images and the requirements specified by $l$ is simply multiplicative. However, consider a more general case where the given classifier $f(x_0)$ corresponds to a very special property and only a tiny proportion of images gives $f(x_0) \approx 1$ (e.g., whether the generated image is Larosterna inca, a bird species native to the coastal regions of western South America). Most unconditional images will not even have activated embeddings of the last layer of $f$ since they have a close-to-zero classifier output. In such case, the property of being able to match the embeddings of $f(x_t,t)$ and $f(x_0)$ is unlikely generalizable to desired images with $f\approx 1$, since both noisy and clean undesired images could have close-to-zero embeddings but both noisy and clean desired images do not. Consequently, an extremely large amount of unconditional generation is required to help learn a time-dependent classifier of one particular property. We are willing to see if future works can give a more quantitative result about the training efficiency for general classifiers.
> >
> > > For Question 3, you assert that Theorem 3.2 represents the main theoretical contribution. However, it appears to be straightforward as the proposed method is a combination of these methods. Could you explain why the proof is non-trivial?
> > >
> >
> > We would like to clarify that the major contribution of the unification algorithm does not lie in the novelty or difficulty of the proof of theorem 3.2, but rather in recognizing and quantifying the importance of unifying different methods in the same hyper-parameter space for training-free guidance. For instance, while different existing methods overlap in certain techniques, their intuition and explanation of each of the technique is different, some of which are even incorrect (for example, [C] unconsciously falls into a “fake” training-free guidance setting, as we have discussed in Appendix A.2). By figuring out a way to construct an algorithm with reasonable hyper-parameter space that can encompass existing methods, we unify different techniques and offer a clean way to study the training-free guidance problem. Theorem 3.2 mainly aims to justify the unification theoretically, and the proof is not technically special.

---

> > > ### Author Response · Authors · 2024-08-09
> > > **Response to the followup questions (cont')**
> > >
> > > > In relation to Question 5, I would be interested in a more detailed discussion on the practical implications of the connections between the Ornstein-Uhlenbeck process and time travel. Specifically, how do these connections yield new guarantees or insights that could potentially enhance the methodology of time travel?
> > > >
> > >
> > > We are willing to give a fine-grained theoretical discussion about the insights behind recurrence. Specifically, let’s define a parameterized step $u_t (x)$ that tries to simulate the transformation from distribution $p_{t+1}$ to $p_t$. Then, assume that up to time step $t+1$, the error of distribution estimation is $err_{t+1}$, e.g., if we use the total variation between the resulting distribution $p^\theta_{t+1}$ and $p_{t+1}$, then $err_{t+1} = TV(p_{t+1}^\theta, p_{t+1})$.
> > >
> > > In such case, consider the following OU process:
> > >
> > > $$
> > > x_t = u_t(x_{t+1}),
> > > $$
> > > $$
> > > x_{t+1} = x_t + \epsilon,
> > > $$
> > >
> > > where the recurrent step is $K$, then informally, the Wasserstein-1 distance between resulting distribution $p^\theta_t$ and the ground truth distribution $p_t$ can be controlled by the following terms:
> > >
> > > $$
> > > c_0 (1-\lambda)^K \text{err}_{t+1} + (K+1)c_1,
> > > $$
> > >
> > > where $c_0, c_1$, and $\lambda \in (0,1)$ are some constants that depend on many variables, including score estimation errors, time step $t$, configurations of Gaussian noise $\epsilon$, and more. Intuitively, the first term implies that the contracted error will shrink as we traverse, thanks to the convergent property of the OU process; the second term is an accumulated error due to inaccurate estimation of the transformation from $t+1$ to $t$. The gradient of the upper bound is
> > >
> > > $$
> > > c_0 \text{err}_{t+1} (1-\lambda)^K \ln (1-\lambda) + c_1,
> > > $$
> > >
> > > which, under some assumptions, is negative when $K$ is small and possible after $K$ increases over some certain point. This could intuitively explain why the recurrence is helpful when we increase $K$ from 1 to some certain values, but will lead to under-qualified samples when $K$ becomes overly large: it’s because that recurrence helps find a balance between the contracted error inherited from previous steps and the accumulated error in this step.
> > >
> > > We are happy to add more discussion to the paper, but we want to emphasize a few issues why we cannot make it a formal theorem. First, notice that we can only control the Wasserstein-1 distance at step $t$ using the total variation distance at step $t+1$, which is not transmissible since W1 distance cannot bound TV distance. Second, even if we demonstrate that the upper bound has a decrease-then-increase pattern, it’s not guaranteed that the actual error follows a similar pattern in theory (although it is in practice). The concrete techniques and discussions about the limitations can be found in [D]. We hope that this can help provide insights into the issue you are concerned about.
> > >
> > > We thank you for your time reviewing our paper and for providing supportive comments and constructive feedback. Please let us know if any further clarifications are needed.
> > >
> > > [C] Song, Jiaming, et al. “Loss-Guided Diffusion Models for Plug-and-Play Controllable Generation”. *Proceedings of the 40th International Conference on Machine Learning*, PMLR 202:32483-32498, 2023.
> > >
> > > [D] Xu, Yilun, et al. "Restart sampling for improving generative processes." Advances in Neural Information Processing Systems 36 (2023): 76806-76838.

---

> ### Comment · Reviewer_Xm6V · 2024-08-09
>
> Thank you for your detailed response. I have adjusted my score to an 8. In particular, **I believe the reproducible benchmark used in this paper will influence the trajectory of future research in training-free guidance**. To ensure a comprehensive comparison, I strongly recommend that you include motion diffusion guidance from the LGD paper (a new application related to diffusion planning) and phase retrieval from the DPS paper (a nonlinear non-NN case) as additional benchmarks in the final manuscript. I have not listed this as a weakness, as I understand that these experiments may not be feasible within the rebuttal period.

---

> > ### Author Response · Authors · 2024-08-09
> > **Thanks for the review**
> >
> > We sincerely appreciate your timeply reply and considerate response about the short experiments window. We promise to add them in the revised paper. Thanks!

---

### Official Review · Reviewer_aTDj · 2024-06-24

**Soundness:** 3
**Presentation:** 3
**Contribution:** 3
**Rating:** 6
**Confidence:** 3

**Summary:**

The authors propose a framework (TFG) for training-free guidance of unconditional diffusion models, enabling their application to conditional generation tasks such as super-resolution, deblurring, etc. via the use of a predictor that evaluates the quality of a clean sample.  TFG, as in related past methods, aim to sample from the conditional distribution of samples defined in Equation 4 without either training a conditional diffusion model or a predictor evaluating the quality of noisy samples as in classifier guidance.  The gradient of the predictor on noisy samples combined with the unconditional score produce the correct conditional score function in Equation 5.  The challenge of training-free methods is to approximate this gradient somehow and a variety of approaches have been previously proposed.  TFG defined in Algorithm 1 is shown to include five such previous training-free methods as special cases for particular hyperparameters, demonstrating that these methods can be viewed and understood in a unified framework.  The authors then propose a procedure to optimize TFG's hyperparameters jointly.  By leveraging the larger design space with optimized hyperparameters, performance gains are observed across 14 tasks, 6 diffusion models, and 38 target predictors compared to the past methods subsumed by TFG.

**Strengths:**

- The TFG framework appears to be a non-trivial, novel incorporation of recent training-free methods and helps contextualize the relationships of these methods to one another.  Training-free guidance of diffusion models is an unsolved problem with many recent papers and placing a portion of this literature into a unified framework is a valuable contribution.
- The framework also led to substantive performance gains on generation validity in the comprehensive benchmarks.  The benchmark evaluations included a good variety of tasks, models, and predictors, including particularly out-of-distribution fine-grained label guidance and molecule property guidance.
- The paper is generally well-written and organized.

**Weaknesses:**

- Some statements and claims are overly broad.  The authors claim all existing training-free approaches fit into their framework which is unlikely.  Many training-free methods have been proposed and the authors could better place their framework in the context of recent literature.  Section 3 Figure 1's demonstration that training-based conditional generation outperforms training-free is unsurprising, given that training-free methods are approximations.
- While hyperparameter search was helpful, the unified design space discussion in Section 4.1 led to limited theoretical insight.  The authors could consider expanding this section to improve their methodological contributions.
- The hyperparameter setting strategy described in Section 4.2, key to getting gains versus past methods, could use more detailed explanation (see questions 2-4 below).
- The benchmark Section 5 decides to not compare generation fidelity in favor of comparing the best algorithm in terms of generation validity.  However, throughout the rest of the paper (e.g. figure 1, 2, and 3) tradeoffs between fidelity and validity are emphasized.  Looking at Table 2, TFG is sometimes but not always the best algorithm in terms of both fidelity and validity simultaneously.  The focus on only validity seems unjustified here.

**Questions:**

1. Why does increasing $N_{recur}$ and $N_{tier}$ eventually hurt performance?
2. In section 4.2, does the "structure analysis" that "increase" is best hold more generally on other tasks than label guidance?
3. The number of Monte Carlo samples in the Implicit Dynamic is set to 1 based on Table 1, but that choice needs more justification.  Does more samples than 4 help?  Why is implicit dynamic helpful with few samples?  This seems contrary to the original motivation of estimating an average.
4. In section 4.2, the beam search in "Searching strategy" could use more detail.  What sample sizes are used and how did that enable quick search? Also, what metric is used when deciding the top K candidates in this search?

**Limitations:**

Limitations are briefly discussed in Section 6, focusing on why training-free guidance remains relevant given language-based image generators.

---

> ### Author Rebuttal · Authors · 2024-08-07
>
> We sincerely thank you for your insightful and constructive review, and we are honored that you think our work is non-trivial, novel, organized, comprehensive, substantive in experiments, and presents a valuable contribution. We are happy to address your concerns:
>
> > The authors claim all existing training-free approaches fit into their framework which is unlikely. Many training-free methods have been proposed and the authors could better place their framework in the context of recent literature. Section 3 Figure 1's demonstration that training-based conditional generation outperforms training-free is unsurprising, given that training-free methods are approximations.
> >
>
> Our algorithm is based on a comprehensive contemporary literature review of more than ten papers (see [1]-[10] below) and on a collection of all of their algorithms (that directly or indirectly use the algorithms we unify), trying to provide an elegant and unified way to study the problem. That said, we completely agree with your suggestion that we never fit all methods, and we are more than happy to explicitly clarify this and discuss our framework in the context of TFG literature.
>
> The purpose of Figure 1 is to emphasize our observation that, unlike what previous works have claimed, training-free guidance is far from being addressed, and all existing methods could even fail on a very easy task. In other words, it helps demonstrate our motivation, rather than to help compare training-free methods with training-based methods. We will clarify this.
>
> > While hyperparameter search was helpful, the unified design space discussion in Section 4.1 led to limited theoretical insight.
> >
>
> Thanks for the suggestion! Currently, we organize Section 4.1 in a way that is “less theoretical” and “more intuitive” to help readers better understand each part of the parameter space without being overwhelmed by theory concepts. That said, the study of space involves Ornstein–Uhlenbeck process, Tweedie’s formula (second order), and probability convergence. We will try to extend our theoretical insights and put more fine-grained discussions in the appendix to ensure both intuition and theory are clearly conveyed to readers.
>
> > The hyperparameter setting strategy… could use more detailed explanation (see questions 2-4 below).
> >
>
> We answer each question separately below.
>
> > The benchmark Section 5 decides to not compare generation fidelity in favor of comparing the best algorithm in terms of generation validity. However, throughout the rest of the paper (e.g. figure 1, 2, and 3) tradeoffs between fidelity and validity are emphasized. Looking at Table 2, TFG is sometimes but not always the best algorithm in terms of both fidelity and validity simultaneously. The focus on only validity seems unjustified here.
> >
>
> We thank you for allowing us to further explain this. We completely agree that there is a trade-off between fidelity and validity, and depending on the user’s requirements, different metrics should be emphasized. The reason why we majorly compare validity is that for each algorithm during beam search, the metric we use to select the best run is exactly the validity (in a held-out set, of course). It’s important that the selection metric and the evaluation metric are the same to avoid unfair and tricky comparisons.
>
> On the other hand, users could also arbitrarily combine fidelity and validity into a new “metric” and conduct beam searches via this metric if they prefer. Our framework does not have any restriction on this, and, unsurprisingly, TFG will still outperform existing methods. We thank you for your question, and we would like to clarify this in the paper.
>
> Additionally, regarding to the questions,
>
> > Why does increasing $N_{recur}$ and $N_{iter}$ eventually hurt performance?
> >
>
> In fact, this phenomenon has been pointed out in previous works (e.g., MPGD) as well, and its underlying theory remains unclear. One observation is that if $N$ is too large, the generated images tend to be “valid” but highly unrealistic, possibly due to the amount of injected noise being too large.
>
> > In section 4.2, does the "structure analysis" that "increase" is best hold more generally on other tasks than label guidance?
> >
>
> Yes. Generally speaking, we find that “increase” works best (or with a negligible gap) among all of the tasks we consider in the paper.
>
> > The number of Monte Carlo samples in the Implicit Dynamic is set to 1 based on Table 1, but that choice needs more justification. Does more samples than 4 help? Why is implicit dynamic helpful with few samples? This seems contrary to the original motivation of estimating an average.
> >
>
> We find that more than $4$ of sample size does not help with sample quality because it reduces the stochasticity of the dynamics (think about changing the Gaussian noise in an SDE to, e.g., its expectation). We are delighted to provide more justification by adding a mathematical explanation of the role that the Gaussian noise is playing in the appendix and explaining why more samples are not beneficial.
>
> > In section 4.2, the beam search in "Searching strategy" could use more detail. What sample sizes are used and how did that enable quick search? Also, what metric is used when deciding the top K candidates in this search?
> >
>
> As mentioned in line 286-287, all searches are run with 1/8 of the test sample size and a maximum search step of 6. The test sample size for each task can be found in Appendix D. To conduct exhaustive grid search, we need to run more than $125$ experiments, which are much slower than our beam search strategy. The metric used when deciding the top K candidates is validity.
>
> Again, we sincerely appreciate your helpful review. Please let us know whether there is any additional concern that we could address to improve your evaluation of our work.

---

> ### Author Response · Authors · 2024-08-07
> **Additional references**
>
> [1] Controllable Music Production with Diffusion Models and Guidance Gradients.
>
> [2] A Framework for Conditional Diffusion Modelling with Applications in Protein Design and Inverse Problems.
>
> [3] Solving Audio Inverse Problems with a Diffusion Model.
>
> [4] Diffusion Models for Audio Restoration.
>
> [5] Vrdmg: Vocal Restoration via Diffusion Posterior Sampling with Multiple Guidance.
>
> [6] Motion Guidance: Diffusion-Based Image Editing with Differentiable Motion Estimators.
>
> [7] Training-free Multi-objective Diffusion Model for 3D Molecule Generation.
>
> [8] Control3diff: Learning Controllable 3D Diffusion Models from Single-view Images.
>
> [9] Steered Diffusion: A Generalized Framework for Plug-and-Play Conditional Image Synthesis.
>
> [10] Contrastive Energy Prediction for Exact Energy-Guided Diffusion Sampling in Offline Reinforcement Learning.

---

> > ### Comment · Reviewer_aTDj · 2024-08-09
> >
> > I thank the authors for their rebuttal and believe the discussed changes may improve the paper upon revision.

---

> > > ### Author Response · Authors · 2024-08-09
> > > **Thanks for your feedback!**
> > >
> > > We thank the reviewer for the timely response. We highly appreciate it the reviewer could increase the evaluation score accordingly if you believe that the revised paper is improved.

---

### Official Review · Reviewer_J9FB · 2024-07-11

**Soundness:** 3
**Presentation:** 3
**Contribution:** 3
**Rating:** 6
**Confidence:** 4

**Summary:**

This paper scrutinizes existing works on training-free guidance in diffusion models and proposes a unified framework that includes all existing methods as special cases. With this unified framework, this work presents a detailed and informed investigation of the design choices and hyperparameters within this framework. Additionally, it proposes a comprehensive benchmark involving 14 task types and 38 targets to evaluate the performance of this unified framework by optimizing the hyperparameters in the design choices.

**Strengths:**

1. The proposed unified framework is a very interesting and novel summarization of the methods in existing works. Existing training-free guidance often uses different notations and different ways to formulate the problem. It is very helpful that this framework elucidates the design choices in training-free guidance.

2. The proposed benchmark is very comprehensive, and I believe it will certainly be helpful for the community to continue conducting more research on this topic.

**Weaknesses:**

1. While I appreciate the comprehensiveness of the experiments on the proposed benchmark, I think the author fails to provide an informative analysis of the results. Currently, the results simply show that by optimizing the hyperparameters in the unified framework, we obtain better performance, which is a natural outcome since the existing methods are included in the framework. From a research perspective, I can think of several questions worth investigating, such as whether the optimal hyperparameters vary between tasks and, when using fixed models, how the optimal hyperparameters are affected by the target objective function. With this investigation, it would be best to reach some general conclusions to guide users in tuning the hyperparameters in practice, rather than just relying on grid search.

2. It seems that for around half of the tasks, the improvement is only marginal, while for a few other settings, the improvement is significant. I think this phenomenon is worth further investigation to understand in what scenarios TFG will provide improvements.

**Questions:**

1. Why does the highest MO energy in Table 3 show a negative improvement? MPGD should be a special case of TFG. Is this because the grid-searched hyperparameters for TFG are not optimal?

2. Can the author comment on other lines of work [1,2,3,4], specifically direct optimization approaches, on training-free optimization of diffusion models with a target objective? The task setting is the same as training-free guidance. Although it is evident that the direct optimization approaches are slower than training-free guidance, it is not clear how their performance compares.

[1] Bram Wallace, Akash Gokul, Stefano Ermon, and Nikhil Naik. End-to-end diffusion latent optimization improves classifier guidance. In Proceedings of the IEEE/CVF International Conference on Computer Vision, pages 7280–7290, 2023b.

[2] Heli Ben-Hamu, Omri Puny, Itai Gat, Brian Karrer, Uriel Singer, and Yaron Lipman. D-flow: Differentiating through flows for controlled generation. arXiv preprint arXiv:2402.14017, 2024.

[3] Korrawe Karunratanakul, Konpat Preechakul, Emre Aksan, Thabo Beeler, Supasorn Suwajanakorn, and Siyu Tang. Optimizing diffusion noise can serve as universal motion priors. arXiv preprint arXiv:2312.11994, 2023.

[4] Tang, Zhiwei, et al. "Tuning-Free Alignment of Diffusion Models with Direct Noise Optimization." arXiv preprint arXiv:2405.18881 (2024).

**Limitations:**

See my comments on the weaknesses above. I think there are a few unclear points worth discussing.

---

> ### Author Rebuttal · Authors · 2024-08-07
>
> We sincerely thanks for your insightful and constructive review, and we are more than honored that you think our work is interesting, novel, comprehensive, and will be helpful for the community. Regarding your concerns,
>
> > While I appreciate the comprehensiveness of the experiments on the proposed benchmark, I think the author fails to provide an informative analysis of the results. Currently, the results simply show that by optimizing the hyperparameters in the unified framework, we obtain better performance, which is a natural outcome since the existing methods are included in the framework. From a research perspective, I can think of several questions worth investigating, such as whether the optimal hyperparameters vary between tasks and, when using fixed models, how the optimal hyperparameters are affected by the target objective function. With this investigation, it would be best to reach some general conclusions to guide users in tuning the hyperparameters in practice, rather than just relying on grid search.
> >
>
> We agree the importance of more fine-grained analysis, and we are delighted to investigate more on our experimental results as you suggested. Specifically, we first explicitly present the optimal hyper-parameters of TFG and other existing methods for all tasks in the uploaded PDF (Table 2). We list several observations below.
>
> 1. Overall, optimal parameters vary widely between problems and datasets. For example, even with the same model and objective (e.g., label classifier on ImageNet or CIFAR10), the best hyperparameters vary widely from target to target. This highlights the importance of hyperparameter search.
> 2. The improvement of TFG over existing methods depends heavily on the difference between the optimal parameters and the subspaces of existing methods. See the next questions for a more detailed analysis.
> 3. Also note that in our paper, we didn’t do an exhaustive grid search, but instead perform an efficient beam search (line 254-264) to find the optimal hyper-parameters. All searches are run with 1/8 of the test sample size and a maximum search step of 6 (line 287). Our experimental results demonstrate that this is sufficient to find near optimal parameters (check out Table 1 in our uploaded PDF).
>
> > It seems that for around half of the tasks, the improvement is only marginal, while for a few other settings, the improvement is significant. I think this phenomenon is worth further investigation to understand in what scenarios TFG will provide improvements.
> >
>
> Thanks for the advice! We conducted a detailed investigation into the phenomenon and found that the improvements are highly related to the differences in the optimal hyperparameters between TFG and existing methods. For example, the $\bar\rho$ for UGD is the same as TFG for gender-age guidance task, where TFG only has 0.133% validity improvement over UGD. On the contrary, their values differ on the fine-grained classification task, and TFG has an 18.7% validity improvement over UGD. Overall, this depends on whether the optimal parameter lies in the subspace that existing methods can find. We will add this analysis to our revised manuscript.
>
> In addition, regarding to the questions,
>
> > Why does the highest MO energy in Table 3 show a negative improvement? MPGD should be a special case of TFG. Is this because the grid-searched hyperparameters for TFG are not optimal?
> >
>
> We appreciate the reviewer for carefully examining our paper, and we want to point out that the reason TFG could occasionally have slightly worse performance in practice is due to the beam search computation limit we currently pose. More specifically, we allow TFG to search at most six steps (for all hyper-parameters) and all other methods for seven steps (in their subspaces). For the MO energy task, the searched parameter for MPGD is that $\bar \mu = 0.016$ (this is the only parameter that we need to search for MPGD), where the best (and last step) of TFG is that $\bar \mu = 0.004$ (because it uses one step to double another parameter). If we allocate more computational budget for the beam search steps, TFG will outperform MPGD on this target (in fact, eight steps suffice).
>
> > Can the author comment on other lines of work [1,2,3,4], specifically direct optimization approaches, on training-free optimization of diffusion models with a target objective? The task setting is the same as training-free guidance. Although it is evident that the direct optimization approaches are slower than training-free guidance, it is not clear how their performance compares.
> >
>
> Thanks for the valuable suggestion. We review all papers to understand the direct noise optimization approach (DNO), and we believe that this DNO has a different motivation than training-free guidance. Specifically, DNO is not only slow but also GPU-memory intensive, as gradients have to be propagated through the entire ODE process for multiple times until convergence. This makes it hard to implement and study within a short amount of time. We would like to leave the comparison to future studies, and we sincerely hope you can understand the difficulty.
>
> Again, we sincerely thanks for the helpful review you provide. Please let us know whether there is any additional concern that we could address to help improve your evaluation to our work.

---

> ### Author Response · Authors · 2024-08-12
> **A gentle reminder**
>
> Dear Reviewer J9FB,
>
> The deadline of the discussion period is soon approaching. We wonder whether our answers to your questions have addressed your concerns. If there are any additional discussion points or questions, we are happy to discuss.
>
> We look forward to your comments. Thank you again for your time.
>
> Best,
>
> Authors of Paper8134

---

### Official Review · Reviewer_DCpp · 2024-07-21

**Soundness:** 3
**Presentation:** 3
**Contribution:** 3
**Rating:** 5
**Confidence:** 3

**Summary:**

This paper focuses on the unification of training-free guidance methods for diffusion models. It defines each method within a unified framework and finds that restricting the hyperparameter space is consistent with existing methods. This framework can be broadly categorized into mean guidance, variance guidance and implicit dynamics. The paper demonstrates performance improvements in several experimental scenarios.

**Strengths:**

* Experiments were conducted on different data sets and scenarios.
* Training-free guidance, which had been developed in different ways, was unified into a single framework.
* If the code is open-sourced, it will greatly benefit the diffusion community.

**Weaknesses:**

* Algorithm 1 seems to be one of the most important parts of this paper, but it lacks a detailed explanation. An explanation of Algorithm 1 along with the corresponding hyperparameters from Definition 3.1 in Section 3.1 would greatly aid understanding.

* In Algorithm 1, the iteration part in line 8 could be explicitly described using a for loop in the algorithm, rather than as a comment.

* I believe that Theorem 3.2 and its proof are not well formulated mathematically. If the authors want to formalize it as a theorem, they need to show mathematically that the same generated distribution can be produced. In my opinion, an explanation at an analytical level would be acceptable.

**Questions:**

Please see the Weaknesses part.

**Limitations:**

They provided in the last section.

---

> ### Author Rebuttal · Authors · 2024-08-07
>
> Thank you for the insightful and constructive review, and we are delighted that you think our work will greatly benefit the diffusion community. Regarding to the concerning questions:
>
> > Algorithm 1 seems to be one of the most important parts of this paper, but it lacks a detailed explanation. An explanation of Algorithm 1 along with the corresponding hyperparameters from Definition 3.1 in Section 3.1 would greatly aid understanding.
> >
>
> We agree that more explanation w.r.t. Alg 1 and the hyperparameter space will help with the understanding. We will incorporate the explanation of the relation between Alg 1 and Def 3.1: generally speaking, Alg 1 contains three operations (Mean Guidance, Variance Guidance, and Implicit Dynamics), and each operation in the algorithm separately control part of the hyper-parameter in Def 3.1. We will explicitly point how this relations, and how each operation in the algorithm affects the parameter space, i.e., what part of the space is controlled.
>
> > In Algorithm 1, the iteration part in line 8 could be explicitly described using a for loop in the algorithm, rather than as a comment.
> >
>
> We agree that explicitly writing down the for loop will make it more clear. Thanks for the suggestion.
>
> > I believe that Theorem 3.2 and its proof are not well formulated mathematically. If the authors want to formalize it as a theorem, they need to show mathematically that the same generated distribution can be produced. In my opinion, an explanation at an analytical level would be acceptable.
> >
>
> We really appreciate the suggestion, and we believe that keeping it as a theorem would still be helpful. That said, we will make the definition and proof more mathematically formal. Specifically, we will explicitly define the parameter space of each of the existing algorithms, and pointing out that each instantiation in their distribution corresponds to which parameter value in our parameter space.
>
> Again, we sincerely thanks for the helpful review you provide. Please let us know whether there is any additional concern that we could address to help improve your evaluation to our work.

---

> > ### Comment · Reviewer_DCpp · 2024-08-13
> >
> > I appreciate the author's response. I expect my concerns are well reflected in the revised version, and I think the current score is appropriate, so I keep it.

---

> ### Author Response · Authors · 2024-08-12
> **A gentle reminder**
>
> Dear Reviewer DCpp,
>
> The deadline of the discussion period is soon approaching. We wonder whether our answers to your questions have addressed your concerns. If there are any additional discussion points or questions, we are happy to discuss.
>
> We look forward to your comments. Thank you again for your time.
>
> Best,
>
> Authors of Paper8134

---

### Author Rebuttal · Authors · 2024-08-07

We sincerely thank all reviewers for their insightful and constructive reviews, and we are honored that all reviewers believe that our paper is novel, beneficial, comprehensive, and well-written. Reviewer aTDj thinks the work is “a valuable contribution”, reviewer Xm6V thinks it “reflects a high degree of diligence and thoroughness”, and both reviewer DCpp and J9FB think it will “certainly be helpful for the community”.

All reviewers provide fine-grained and important feedback on polishing the paper, and we are more than willing to accept. As NeurIPS policy does not allow us to upload an improved version, we will explicitly explain how we improve the paper in each of your questions, and we promise that all suggestions will be addressed in the final paper. As all reviewers have emphasized the importance of the codebase and benchmarks, we will open-source our code, benchmarks, configurations, and existing runs, as promised in the paper.

Reviewer aTDj points out that the claim on unifying *all* works is overly broad, and indeed, reviewers J9FB and Xm6V list a few training-free guidance papers that are related but with different focus. We want to highlight that our work is based on a comprehensive contemporary literature review over more than 15 papers (see response to reviewer aTDj) and on a collection of all of their algorithms (that directly or indirectly use the algorithms we unify), trying to provide an elegant and unified way to study the problem. That said, we completely agree that we never fit all methods, and we are more than happy to explicitly clarify this and discuss our framework in the context of TFG literature.

For the uploaded PDF, we present the actual parameter selected by our beam search strategy for each algorithm and task, and we additionally compare the performance of the beam search parameter with the full grid search one. Overall, these results demonstrate that the comparison between different methods is transparent and objective, and that our beam search strategy is effective and efficient compared with full grid search method.

Below are our detailed responses. Please do not hesitate to let us know if there is any additional concern that we can help address to help with an objective evaluation of our work.

---

### Decision · Program_Chairs · 2024-09-25

**Decision:**

Accept (spotlight)

**Comment:**

This paper addresses the challenge of generating samples with desired target properties using an unconditional diffusion model and a target property predictor without requiring additional training. It critiques existing methods for lacking theoretical grounding and rigorous benchmarking, which can lead to inconsistent performance across tasks. The authors introduce a novel algorithmic framework that unifies existing methods under a broader, algorithm-agnostic design space. They also propose an efficient hyper-parameter search strategy and demonstrate its effectiveness through extensive benchmarking, showing a 7.4% average performance improvement across various models and tasks. The reviewers have been generally positive about the contribution of this work, and believe that the proposed benchmarking will be interesting to future works in the related areas.